# Phosphoproteomic-based kinase profiling early in influenza virus infection identifies GRK2 as antiviral drug target

Emilio Yánguez[1], Annika Hunziker[1], Maria Pamela Dobay [2], Soner Yildiz[3], Simon Schading[1], Elizaveta Elshina[1], Umut Karakus[1], Peter Gehrig[4], Jonas Grossmann[4], Ronald Dijkman [5,6], Mirco Schmolke[3] & Silke Stertz[1]

Although annual influenza epidemics affect around 10% of the global population, current treatment options are limited and development of new antivirals is needed. Here, using quantitative phosphoproteomics, we reveal the unique phosphoproteome dynamics that occur in the host cell within minutes of influenza A virus (IAV) infection. We uncover cellular kinases required for the observed signaling pattern and find that inhibition of selected candidates, such as the G protein-coupled receptor kinase 2 (GRK2), leads to decreased IAV replication. As GRK2 has emerged as drug target in heart disease, we focus on its role in IAV infection and show that it is required for viral uncoating. Replication of seasonal and pandemic IAVs is severely decreased by specific GRK2 inhibitors in primary human airway cultures and in mice. Our study reveals the IAV-induced changes to the cellular phosphoproteome and identifies GRK2 as crucial node of the kinase network that enables IAV replication.

---

[1] Institute of Medical Virology, Winterthurerstrasse 190, 8057 Zurich, Switzerland. [2] Bioinformatics Core Facility, SIB Swiss Institute of Bioinformatics, Quartier Sorge—Batiment Genopode, Lausanne, 1015, Switzerland. [3] Department of Microbiology and Molecular Medicine, University of Geneva, Rue Michel-Servet 1, CH-1211 Geneva, Switzerland. [4] Functional Genomics Centre Zurich, Winterthurerstrasse 190, 8057 Zurich, Switzerland. [5] Institute of Virology and Immunology, Länggassstrasse 122, 3012 Bern, Switzerland. [6] Department of Infectious Diseases and Pathobiology, Vetsuisse Faculty, University of Bern, Bern, Switzerland. These authors contributed equally: Annika Hunziker, Maria Pamela Dobay, Soner Yildiz. Correspondence and requests for materials should be addressed to S.S. (email: stertz.silke@virology.uzh.ch)

Influenza A viruses (IAV) still pose a substantial burden on human health and worldwide economics. Seasonal influenza viruses are responsible for up to 500,000 deaths annually, with immunocompromised individuals at particularly high risk for severe courses of infection. The appearance and transmission of pandemic IAV strains, which have caused devastating outbreaks in the past, additionally threatens global health and urges the discovery of new antivirals. Cellular factors involved in viral replication have been proposed to be attractive targets for anti-viral development[1–3]. Among them, kinases are particularly promising, as kinase inhibitors comprise up to 30% of drug-discovery programs in the pharmaceutical industry[3,4].

IAV harnesses the cellular endocytic machinery to enter the cell and traffic through the cytoplasm to reach the replication site in the nucleus. Coordinated early activation of signaling pathways has been shown to be important for viral entry[5–13] and identification of key kinases involved in this process could contribute to the development of new antivirals. Binding of IAV particles, by interaction of the viral hemagglutinin (HA) to exposed sialylated proteins on epithelial cells[14], has been proposed to induce the formation of lipid raft-based signaling platforms, in which receptor tyrosine kinases (RTKs) such as the epidermal growth factor receptor (EGFR) or c-Met, are activated[6]. Clustering of activated RTKs leads to their internalization in endocytic vesicles, in which the viral particles could be engulfed[15]. Downstream of this initial RTK-signaling, early activation of the phosphatidylinositol-3 kinase (PI3K) has been shown to promote IAV endocytosis[5–7] and, together with the extracellular signal-regulated kinase ERK1/2, to enhance the activity of the vacuolar-type $H^+$-ATPases (vATPases)[8,16], which are essential for endosomal acidification leading to viral fusion[17–19]. Focal adhesion kinase (FAK) has been proposed to establish a link between this PI3K activation and the cytoskeleton reorganization required for viral endosomal trafficking[9] and the activation of protein kinase C (PKC) has been shown to play a role in IAV trafficking through late endosomes[10,11]. More recently, $Ca^{2+}$ signaling has also been implicated in both, clathrin-dependent and clathrin-independent IAV entry mechanisms via an intricate associated regulatory network[12].

However, a systematic and unbiased analysis of the main signaling routes initiated by IAV binding and key mediators required for subsequent infection is still lacking. Here we conduct a SILAC-based quantitative phosphoproteomic analysis of human lung epithelial cells within minutes post-infection. We quantify the phosphorylation status of around 3000 different phosphorylation sites from >1300 proteins and identify infection-induced changes in the phosphorylation pattern. On the basis of this virus-induced phospho-signature, we are able to identify kinases, such as the G protein-coupled receptor kinase 2 (GRK2), that are activated during IAV entry and responsible for the observed signaling landscape. Inhibition of GRK2 kinase activity severely decreases IAV uncoating and inhibits viral replication in primary human airway epithelial cultures, as well as in an animal model of IAV pathogenesis. Our results therefore establish GRK2 as a promising drug target for the next generation of antivirals for influenza virus.

## Results

### IAV entry induces a unique phosphorylation signature. In order to identify cellular kinases required for IAV entry into cells, we conducted a quantitative phosphoproteomic screen on A549 human lung epithelial cells. We hypothesized that virus binding to host cells would already induce signaling cascades that enable the following steps of the replication cycle. As tyrosine phosphorylation of epidermal growth factor receptor (EGFR) had

been shown to be induced by HA binding to host cells[6], we monitored EGFR phosphorylation upon infection of A549 cells with IAV strain A/WSN/33 (MOI = 25 PFU/cell). We observed strong activation of EGFR at 5 and 15 min post infection (p.i.), and therefore selected these time points for our analysis (Supplementary Figure 1a). For accurate quantification of phosphorylation dynamics, we performed five biological replicates in A549 cells subjected to triple isotope labeling by amino acids in cell culture (SILAC)[20], from which the peptides phosphorylated at serine or threonine residues were enriched, and identified and quantified by liquid chromatography coupled to tandem mass spectrometry (LC-MS/MS) (Fig. 1a). We used IAV A/WSN/33 grown in embryonated chicken eggs for our analysis to avoid contributions from cytokines, such as interferons, as these act mostly in a species-specific manner. In addition, we also performed a proteomic analysis for chicken proteins present in the virus stock and the allantoic fluid from mock-infected eggs that we used for our mock condition (Supplementary Data 1). We did not detect any chicken protein that was at least 2-fold more abundant compared to the mock, allowing us to attribute differences in phosphorylation to virus infection. We were able to detect >3000 different phosphosites per experimental condition, from around 2400 phosphopeptides belonging to about 1300 proteins (Supplementary Figure 1b, raw data in Supplementary Data 2). As expected, the vast majority of the detected phosphosites were phospho-serine and -threonine modifications, and only few phospho-tyrosines. A total of 70% of the phosphosites were detected in more than one biological replicate with a good correlation between them (Supplementary Figure 1c, d). Quantitative analysis performed using MaxQuant[21] revealed changes in phosphorylation of an important subset of phosphopeptides within minutes of virus addition, while the absolute peptide abundance, quantified from total extract digests, remained mostly unaltered. Using a cutoff value of 1.5-fold change (FC) compared to mock infected samples, 213 phosphosites were found to be differentially phosphorylated 5 min post-inoculation (Fig. 1b), 133 of which were dephosphorylated and 80 showing an increase in phosphorylation. 15 min after the addition of virus, changes in phosphorylation were more pronounced, with 375 sites being differentially phosphorylated (Fig. 1c), 257 dephosphorylated and 118 showing an increase in phosphorylation. These data reveal that different cellular signaling pathways are activated within minutes of IAV infection, possibly induced by virus binding to the cell surface.

To determine which cellular functions are regulated by phosphorylation early in IAV infection, we analyzed the cellular pathways to which the proteins containing IAV-responsive phosphosites are assigned. The distributions of annotated KEGG (Kyoto Encyclopaedia of Genes and Genomes) pathways[22] among differentially phosphorylated proteins ($\log_2 FC \geq 0.5$, at either 5 or 15 min post-infection) were compared against the background of the total population of identified phosphorylated proteins (Fig. 1d and Supplementary Table 1). Proteins with a described function in MAPK signaling (adj. $p$ value = $2.29e^{-08}$ (hypergeometric test)), endocytosis (adj. $p$ value = $1.13e^{-06}$ (hypergeometric test)) and regulation of the actin cytoskeleton (adj. $p$ value = $5.79e^{-06}$ (hypergeometric test)) were overrepresented among the proteins phosphorylated in response to infection. As these cellular functions are known to be required for IAV entry, this result validates our experimental approach[5–8,16,23–26]. Furthermore, proteins related to focal adhesions and tight junctions (adj. $p$ value = $6.47e^{-03}$ (hypergeometric test)), thus far not described to be involved in the early stages of IAV infection, were found to be significantly enriched. Among them, integrins (ITGB4), actin-binding proteins such as parvins (PARVA) and filamins (FLNB), proteins related to actin-membrane attachment (PXN, EPB41L1, VAPA) and different

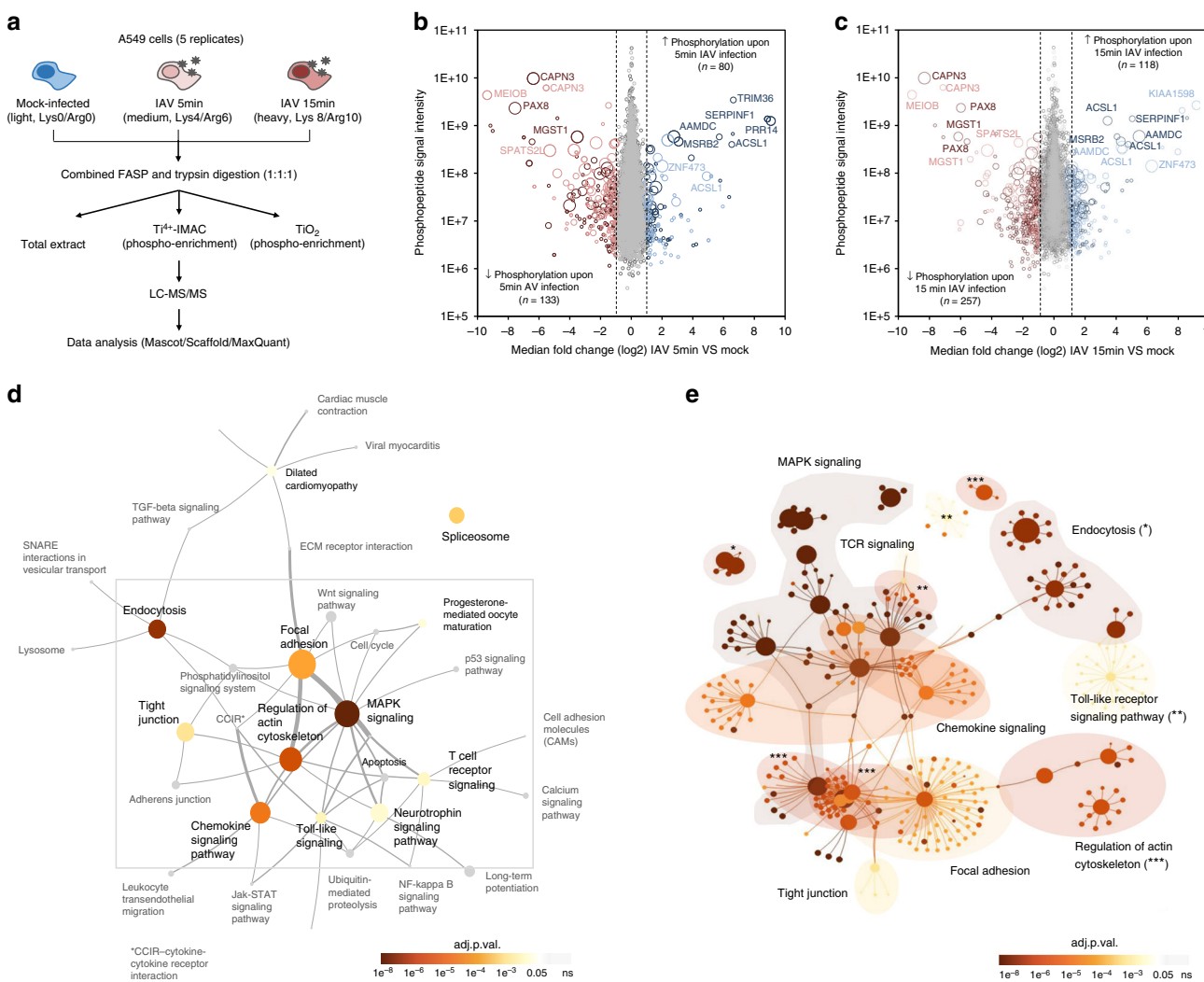

**Fig. 1** IAV-induced changes in the host cell phosphoproteome. **a** Overview of the experimental design. **b**, **c** Distribution of changes in cellular phosphorylation after 5 (**b**) and 15 (**c**) min of infection of A549 cells with IAV strain A/WSN/33 at MOI = 25 PFU/cell. Median fold-changes in phosphorylation of the identified phosphosites are presented in relation to the intensity of detection. Dephosphorylated peptides are shown in red and hyper-phosphorylated ones in blue. Darker colors are used for the results of the $Ti^{+4}$ chromatography, lighter colors for $TiO_2$. The size of the circles reflects the number of biological replicates in which a given peptide was identified. The names of the proteins for the main differentially phosphorylated peptides are indicated. **d** The main IAV-responsive pathways are represented by circles in the context of the KEGG pathway topology. The size of the circles reflects the number of differentially phosphorylated peptides, while the color indicates the significance of enrichment from the standard hypergeometric test. Edge width corresponds to the number of shared components between a pair of KEGG networks. **e** The differentially phosphorylated proteins in the top enriched pathways are represented in the context of a functional association network. Asterisks are used to identify small or distant nodes for the represented pathways. Vertices are colored based on the adjusted $p$-values for enrichment from the standard hypergeometric test

cytoskeleton components (TLN1), which were not previously associated with viral entry, were phosphorylated within minutes of infection. We also generated a network to analyze how the differentially phosphorylated proteins are interconnected within the different pathways and identified a clear cross-talk between them (Fig. 1e).

**Kinase profiling reveals IAV-responsive kinases**. As our main goal was to reveal novel drug targets for IAV, we next focused on the kinases activated in response to infection. To this aim, we developed a bioinformatics pipeline to identify cellular kinases that could be responsible for the phosphorylation of the IAV-responsive phosphosites in our data set. Our approach centres on the kinase predictions obtained from the Group-based Prediction System 3.0 (GPS 3.0)[27–29] including a redefined high stringency cutoff (see Methods and Supplementary Figure 2). GPS 3.0 ranks the likelihood that a particular kinase or kinase family

phosphorylates a given residue considering the amino acids surrounding the phosphorylation site. Putative kinases for the hyperphosphorylated sites at both 5 and 15 min post-infection ($log_2FC \geq 0.5$) were identified and the distribution of the top predicted IAV-responsive kinases was compared against the background of the kinases predicted for the total population of identified phosphopeptides (Fig. 2a). Putative substrates for mitogen-activated protein kinase 1 (MAP2K1/MEK1), MAP kinase-interacting serine/threonine-protein kinase 2 (MNK2, $p$ value = 0.043 (Fisher's exact test)) and RP6KB1 ($p$ value = 0.023 (Fisher's exact test)) were the top hits among the hyperphosphorylated peptides 5 min after IAV binding. Predicted substrates for cyclin-dependent kinase 1 (CDK1), CDK5 and polo-like kinase 2 (PLK2, $p$ value = 0.003 (Fisher's exact test)) are significantly phosphorylated 15 min after the addition of the virus. Interestingly, G protein-coupled receptor kinase 2 (GRK2), p21 (RAC1) activated kinase 1 (PAK1), Rho-associated coiled-

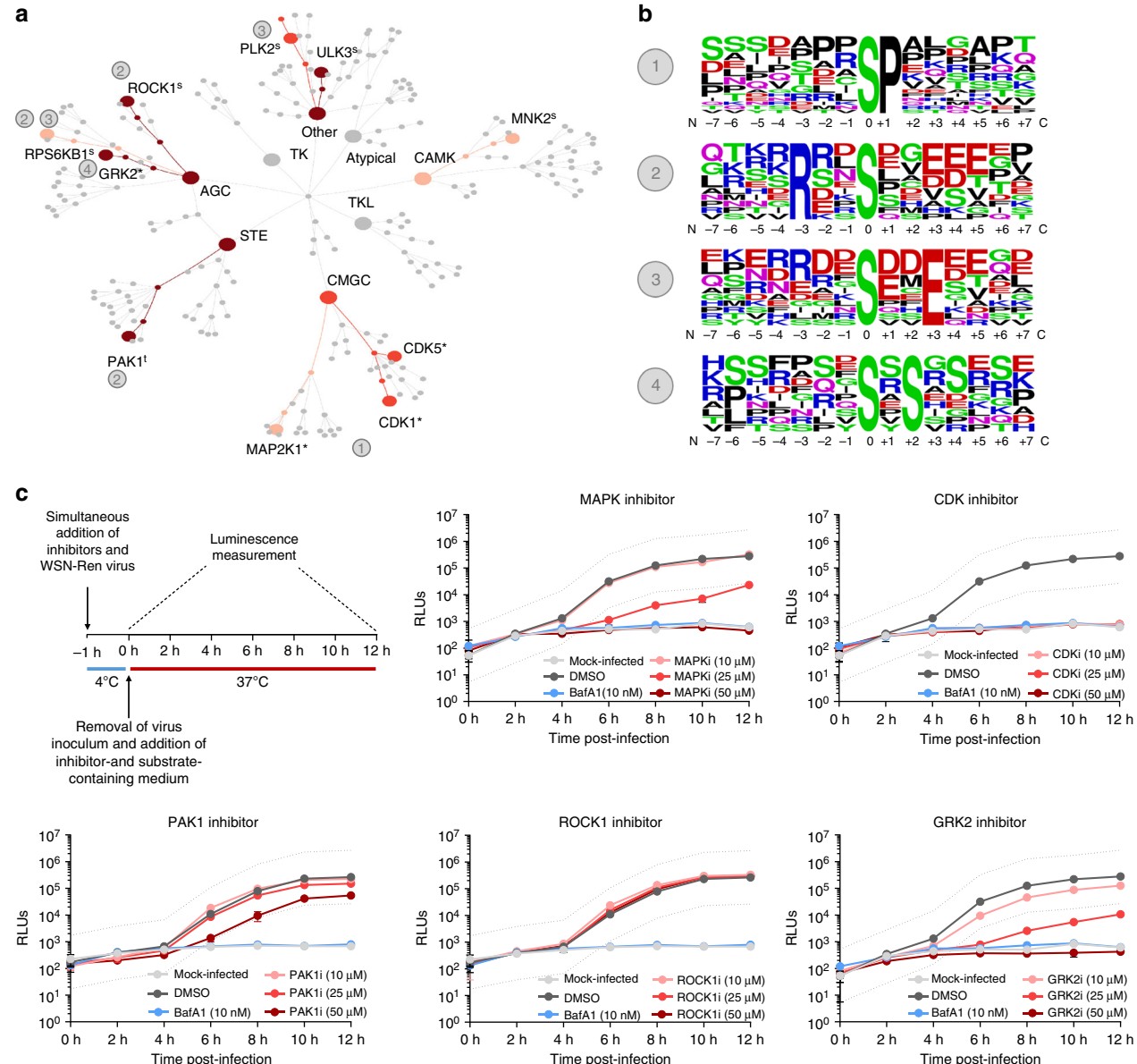

**Fig. 2** Kinase activation profiling reveals IAV-responsive kinases as potential drug targets. **a** Top kinases predicted to be involved in the differential phosphorylation of proteins at 5 and/or 15 min post IAV-infection are shown in color (light orange: predicted to be activated at 5 min p.i., red: predicted to be activated at 15 min p.i.; dark red: predicted to be activated at both time points) in the context of all cellular kinases (s significantly enriched hit, t trend towards significance, * top predicted kinase for which enrichment calculation is not possible, see Methods and Supplementary Fig. 3). **b** Over-represented sequences in IAV-responsive peptides are shown and linked with numbers to the kinases identified in (**a**). **c** Effect of inhibitors against the identified IAV-responsive kinases on the replication of a Renilla-encoding virus. The efficiency of viral replication was determined by luminescence measurement (RLUs) in live cells every 2 h. The dotted line represents a 10-fold difference in viral replication with regard to DMSO-treated cells. Error bars represent standard deviation from three replicates

coil containing protein kinase 1 (ROCK1, *p* value = 0.003 (Fisher's exact test)) and unc-51 like kinase 3 (ULK3, *p* value = 0.002 (Fisher's exact test)) are predicted to be activated at both time-points analyzed. Some of these kinases, like MAPKs or MNKs (MAPK/ERK signaling)[7,8,16], ROCK1[12,23], or PAK1[13,30], have already been implicated in viral entry, and thus validate our analysis. However, we also revealed CDKs, GRK2 and ULK3 as novel kinases activated in early steps of IAV replication. Next, the amino acids adjacent to the hyperphosphorylated positions at 5 and at 15 min post-infection ($\log_2$FC ≥ 0.5) were analyzed to identify conserved sequences that could be linked to the recognition motifs for the predicted kinases. We identified four

sequences enriched among the IAV-responsive phosphopeptides (Fig. 2b). A proline (P) in position +1 is particularly over-represented (39%) among the putative targets for CDKs and MAPK. In agreement with these results, this motif is part of the optimal recognition sequences proposed for CDKs and for ERK and MAPK families[31], whose activation has been shown to be required for IAV entry[6–8,16]. An arginine (R) in position −3 is also enriched among the IAV-responsive phosphopeptides (18%), especially among putative targets for PAK1, ROCK1 and RP6KB1. This particular feature is part of the optimal recognition sequence predicted for PKCs, PKA or ROCK1[31], and these kinases have been linked to IAV entry into cells[10–12,23]. Finally,

a glutamic acid (E) in position +3 (22%) or a serine (S) in position +2 (16%) are likewise associated with IAV-responsive phosphosites among putative targets of PLK2, RP6KB1 or GRK2 in our analysis.

**Inhibition of GRK2 potently inhibits IAV replication**. Next, we evaluated whether viral replication could be reduced by targeting the in silico-identified IAV-responsive kinases with small molecular weight compound inhibitors (Fig. 2c). We tested the effect of specific PAK1 (IPA-3), ROCK1 (Y-27632) and GRK2 (Methyl 5-[2-(5-nitro-2-furyl)vinyl]-2-furoate, GRK2i)[32] inhibitors on the replication of a luciferase-encoding reporter IAV. Together with ULK3, these are the kinases predicted to be activated by IAV at both time-points that were analyzed (Fig. 2a). As inhibitors for ULK3 are not commercially available, ULK3 was excluded from the analysis. We also included a CDK inhibitor (AT7519)[33] because CDKs are key cellular kinases that have not been linked to early steps of IAV infection. Given that the MAPK pathway has been broadly implicated in viral entry[6–8,16] and is among the top hits in our pathway enrichment analysis (Fig. 1d), a specific ERK inhibitor (Ulixertinib)[34] was also included as positive control. Cytotoxicity of the different compounds in A549 cells was determined (Supplementary Figure 3a) and concentrations without deleterious effects were used for further experiments. As expected, the ERK inhibitor efficiently blocked virus replication in a dose-dependent manner (Fig. 2c). PAK1 inhibition partially reduced viral replication, which is consistent with a previous report on the role of PAK1 in IAV infection[13]. We could not observe a significant effect upon ROCK1 targeting in our experimental system but both the CDK and GRK2 inhibitors efficiently blocked viral replication, confirming the prediction of our kinase activation profiling (Fig. 2c). In these cases, the inhibition of viral replication was clearly observed at 4 h post-infection (hpi), suggesting that, as predicted by our phosphoproteomic data, both CDKs and GRK2 are required for an early step in the viral replication cycle.

In order to further explore the possibility of targeting GRK2 to inhibit IAV, we first assessed the impact of GRK2 depletion on viral growth. A549 cells were transfected with four different small interfering RNAs (siRNAs) targeting GRK2, a scrambled siRNA (siSCR) control or an siRNA targeting a vATPase required for endosomal acidification during viral entry. First, silencing efficiency (Supplementary Figure 3b) and impact on cell viability (Supplementary Figure 3c) of the different siRNAs were analyzed. All four siRNAs reduced GRK2 expression to comparable levels but had only minor effects on cell viability. At 48 h post-siRNA transfection, cells were infected with IAV (A/WSN/33) at a low multiplicity of infection (MOI = 0.001 PFU/cell) and supernatants were harvested 24 h and 48 h later (Fig. 3a). Each of the siRNAs reduced viral titers by 10–100-fold, further supporting the role of GRK2 as a host factor required for IAV infection. Next, we performed similar experiments in siRNA-treated A549 cells using the human IAV strain A/Udorn/72 (H3N2) (Fig. 3b) or the highly pathogenic avian strain A/FPV/Dobson/34 (H7N7) (Fig. 3c). GRK2 depletion also inhibited replication of these viruses. Finally, to discard a cell-line-specific requirement, we also transfected siRNAs targeting GRK2 and the above described controls into MRC5 or Wi38 human lung fibroblasts, which were subsequently infected with A/WSN/33 (Supplementary Figure 3d, e). Although the virus was not able to replicate as efficiently as in A549 cells, depletion of GRK2 significantly impaired viral replication in both cell lines.

**IAV induces activation of GRK2**. As GRK2 was one of the top hits predicted to be responsible for the observed IAV-induced phosphorylation signature, we hypothesized that GRK2 was activated by IAV. Previous reports showed that differential phosphorylation regulates GRK2 activity[35–38] and we thus speculated that the phosphorylation status of GRK2 would change in response to IAV infection. However, we had not detected phosphopeptides derived from GRK2 in our proteomic analysis, possibly suggesting activation of GRK2 by tyrosine phosphorylation. In order to assess the phosphorylation status of GRK2 in response to IAV we employed phosphate affinity polyacrylamide gel electrophoresis[39]. This method makes use of a compound that binds to phosphate groups and thereby specifically slows the migration of phosphorylated proteins during gel electrophoresis, which enables visualization of differentially phosphorylated forms of a given protein. As EGF had been reported to induce tyrosine phosphorylation of GRK2, we validated the method by transfecting A549 cells with constructs encoding GRK2 and EGFR, treated cells with epidermal growth factor (EGF) for 5 or 15 min and performed phosphate affinity polyacrylamide gel electrophoresis[35]. As control, we treated the lysates with lambda protein phosphatase (LPP). When cells were stimulated with EGF, we observed a stronger signal for the phosphorylated version of GRK2, which was not present when the samples had been treated with LPP (Fig. 3d). Importantly, we also observed the appearance of the phosphorylated form of GRK2 upon IAV infection (Fig. 3e) confirming our hypothesis that GRK2 is phosphorylated within minutes of infection.

As activation of kinases often leads to their translocation, we also analyzed GRK2 localization in response to IAV infection. We again used EGF treatment as positive control as it had been shown that GRK2 translocates to EGFR-positive clusters at the plasma membrane upon EGFR stimulation[35]. Interestingly, we found that IAV infection also induced the translocation of GRK2 to plasma membrane clusters, where it colocalized with EGFR (Fig. 3f; Supplementary Figure 3f). The translocation of GRK2 was also observed when using a purified stock of IAV (Supplementary Figure 3g), allowing us to attribute GRK2 activation to viral particles and ruling out contributions from other allantoic fluid components. Of note, the IAV-induced GRK2 translocation was dependent on EGFR, as we could not observe it in cells lacking EGFR expression (Fig. 3g). Thus, we conclude that IAV infection induces GRK2 phosphorylation and translocation to EGFR-positive membrane clusters.

**GRK2 kinase activity is required for IAV uncoating**. From our results thus far, we hypothesized that the kinase activity of GRK2 is required for early steps of the viral replication cycle. To verify this, we performed a time of addition experiment in which the GRK2 inhibitor (GRK2i) was added to cells at different times post-infection (Fig. 4a). Addition of the compound together with the virus inoculum efficiently reduced the replication of a Renilla luciferase-encoding reporter IAV. Adding the inhibitor at 2 or 4 hpi reduced viral replication less efficiently, while addition of the inhibitor at later time points did not affect the progression of the infection. Furthermore, addition of the compound only during virus binding or during the first 1–3 h of infection, followed by a washing step and the replacement by inhibitor-free medium, was sufficient to block the replication of the reporter virus (Fig. 4b). To further verify these observations, we performed a similar experiment, using the IAV A/WSN/33 strain, in which supernatants from inhibitor-treated and infected A549 cells were collected 12 hpi and virus titers were determined by plaque assay (Fig. 4c). Addition of the inhibitor during the first 3 h of infection was sufficient to reduce viral production by more than 1000-fold,

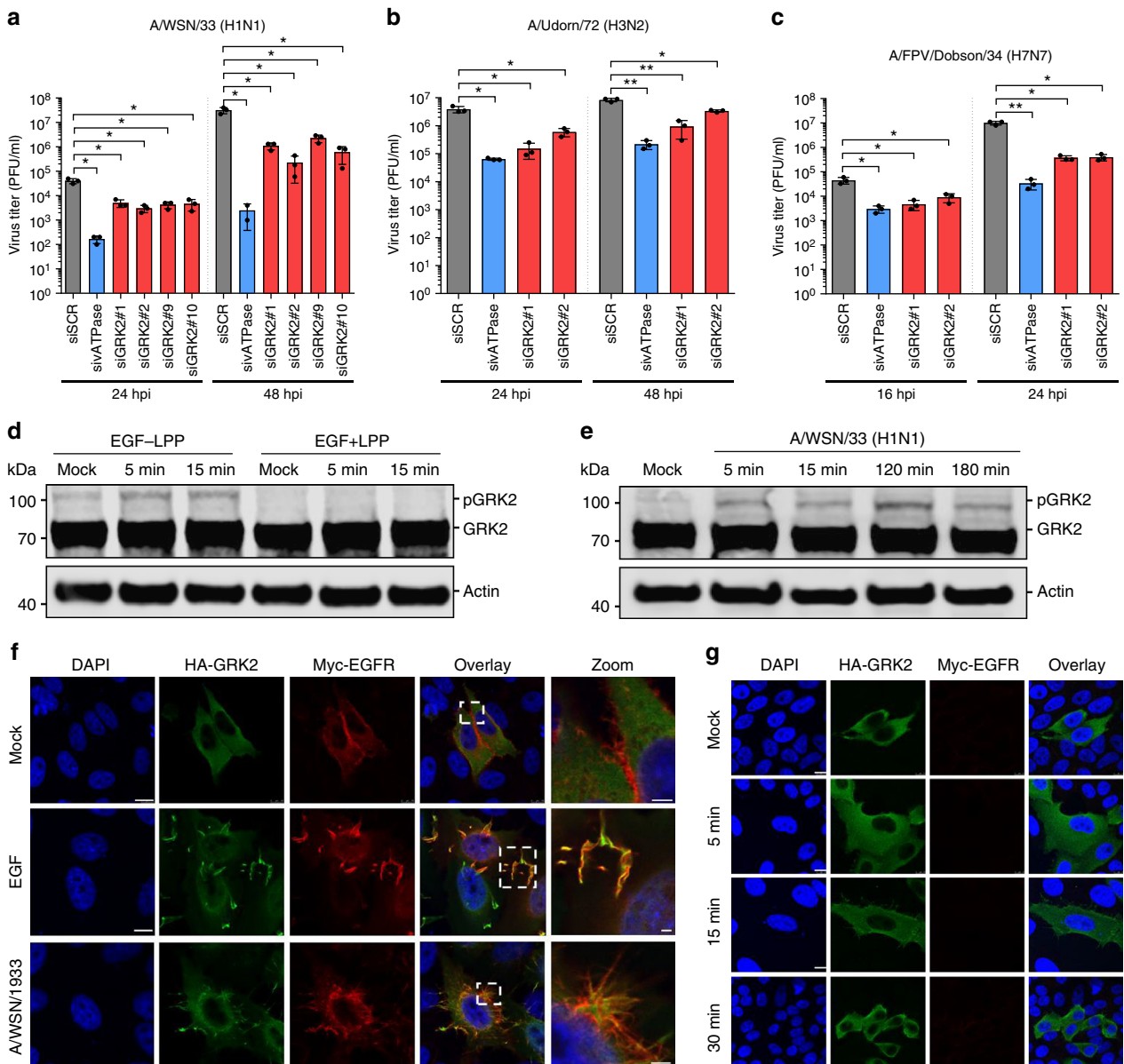

**Fig. 3** GRK2 is a proviral host factor that becomes activated within minutes of IAV infection. **a–c** Effect of GRK2 silencing on the replication of A/WSN/33 (H1N1) (**a**), A/Udorn/72 (H3N2) (**b**) or A/FPV/Dobson/34 (H7N7) (**c**) in A549 cells. At 48 h post siRNA transfection, cells were infected with IAV, supernatants were collected at the indicated times and virus titers were determined by plaque assay. Error bars represent standard deviation from three independent experiments, each performed in triplicates. Statistical significance was determined by unpaired $t$-test. For all panels, ns (non-significant) = $P >$ 0.05; *$P \leq 0.05$; **$P \leq 0.01$. **d** A549 cells were transfected with HA-GRK2 and myc-EGFR-encoding plasmids, serum starved for 16 h before being stimulated with recombinant human epidermal growth factor (hrEGF 100 ng/ml). The phosphorylation status of the indicated proteins was examined using a $Zn^{2+}$ Phos-tag gel. Samples on the right (+LPP) were treated with lambda protein phosphatase (LPP) for 30 min at 30 °C before being run on a $Zn^{2+}$ Phos-tag gel. **e** Same experimental set up as in panel (**d**), but cells were infected with A/WSN/1933 with an MOI = 100 PFU/cell. **f** Hep-2 cells were transfected with HA-GRK2- and myc-EGFR-encoding plasmids and 24 h later stimulated with hrEGF or infected with A/WSN/1933 (MOI = 100 PFU/cell) for 15 min. Cells were fixed and stained using antibodies against the HA-tag (green), EGFR (red) and DAPI (blue) to mark the nuclei. Scale bar corresponds to 10 μm, for zoom images scale bar corresponds to 2.5 μm. **g** Hep-2 cells were transfected with an HA-GRK2-encoding plasmid and 24 h later infected with A/WSN/1933 (MOI = 100 PFU/cell) for 5, 15 or 30 min. Cells were fixed and stained as in (**f**). Scale bar corresponds to 10 μm

confirming the requirement for GRK2 activity during the early steps of the viral cycle. Next, we evaluated the amount of viral nucleoprotein (NP) within the cell nucleus 3 hpi as a direct measurement of the early infection progression. A549 cells were pre-treated for 1 h with GRK2i, subsequently infected with A/WSN/33 at a high multiplicity of infection (MOI = 5 PFU/cell) for 3 h in the presence of the inhibitor and analyzed by confocal

microscopy (Fig. 4d). In control-treated cells, NP was clearly detected in the nucleus as a consequence of ongoing viral replication. However, both GRK2i and BafA1, which was used as positive control, efficiently blocked nuclear NP accumulation. Similar results were obtained in A549 cells transfected with siR-NAs targeting GRK2 (Supplementary Figure 4a). When translation was blocked in a similar experimental setup by addition of

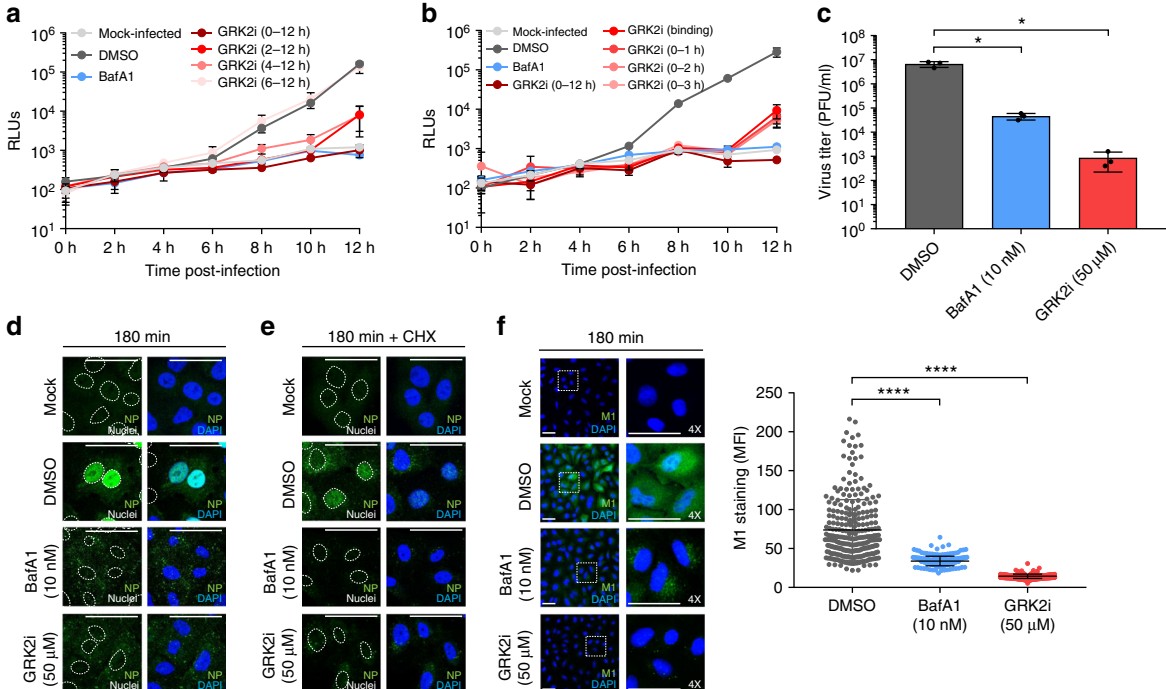

**Fig. 4** GRK2 is required for efficient virus uncoating. **a** A549 cells were infected on ice with a recombinant IAV encoding luciferase (MOI = 1 PFU/cell). At the indicated times post-infection, GRK2i was added to the cells at a final concentration of 50 μM. The efficiency of viral replication was determined by luminescence measurement in live cells every 2 h. **b** A549 were infected and processed as described in (**a**) but GRK2i was added during virus binding or during the first 1 to 3 h of infection, followed by a washing step and the replacement by inhibitor-free medium. One representative out of three independent experiments, each performed in triplicates, is shown in (**a**, **b**). **c** A549 cells were infected with A/WSN/33 (MOI = 1 PFU/cell) and treated with 50 μM GRK2i, 10 nM BafA1 or control medium for 3 h, after which they were incubated in inhibitor-free medium. At 12 hpi, supernatants were collected and virus titers were determined by plaque assay. Mean values from three independent experiments are depicted. **d** A549 cells were treated with the indicated concentrations of the different compounds for 1 h and infected on ice with IAV A/WSN/33 (H1N1, MOI = 5 PFU/cell) for 1 h to synchronize infection. Cells were incubated at 37 °C for 3 h in the presence of the compounds, fixed and stained for viral nucleoprotein (NP, green) and nuclei (blue). Representative images are shown (scale bar corresponds to 40 μm). The white lines indicate the position of the nucleus. **e** Cells were infected and processed as in (**d**) but in the presence of cycloheximide (100 μg/ml). Representative images from three independent experiments are shown (**d**, **e**). **f** A549 cells were treated with the indicated concentrations of the different compounds for 1 h and infected on ice with IAV A/WSN/33 (H1N1, MOI = 25 PFU/cell) for 1 h to synchronize infection. Cells were incubated at 37 °C for 3 h in presence of compounds, fixed and stained for viral M1 protein (M1, green) and nuclei (blue). Representative images are shown (scale bar corresponds to 40 μm). The mean fluorescence green intensity (MFI) in each cell (n > 200) was quantified using ImageJ. One representative out of three independent experiments is shown. For all panels, error bars represent standard deviation (from >200 cells) and statistical significance was determined by unpaired t-test (ns (non-significant) = $P > 0.05$; *$P \leq 0.05$; **$P \leq 0.01$; ***$P \leq 0.001$; ****$P \leq 0.0001$)

cycloheximide in order to detect only incoming viral ribonucleoproteins (vRNPs), the nuclear accumulation of NP was severely impaired in the presence of GRK2i (Fig. 4e), suggesting that the activity of the kinase is required before or during the nuclear import of parental vRNPs.

We then tested whether GRK2 was required for virus internalization or endosomal trafficking. A549 cells were pre-treated for 1 h with GRK2i or equivalent amounts of the solvent dimethyl sulfoxide (DMSO), prior to infection with A/WSN/33 for 30 min at an MOI = 25 PFU/cell. To quantify the amount of virions in early endosomes, we performed confocal immunofluorescence analysis of NP and the early endosome antigen 1 (EEA1) and measured the percentage of co-localization using Imaris software[40,41]. No significant differences in co-localization of internalized virions with EEA1 were observed in cells treated with the inhibitor (Supplementary Figure 4b), suggesting that GRK2 is not required for binding, internalization or early endosomal trafficking. Next, we characterized whether GRK2 function was necessary for efficient viral fusion. To do this, we generated IAV virus-like particles (VLPs) that contain HA and NA from the A/WSN/33 strain and harbor a beta-lactamase reporter protein

fused to the influenza matrix protein-1 (BlaM1)[42]. Upon fusion of viral and endosomal membranes, BlaM1 gains access to the cytoplasm, where it cleaves the fluorogenic substrate CCF-2, which then shifts to a shorter fluorescent emission wavelength that can be detected by flow cytometry. Although GRK2 impairment can efficiently block nuclear accumulation of NP in MDCK cells 3 hpi at non-toxic concentrations of GRKi (Supplementary Figure 4c, d), no differences in viral fusion were detected in the presence of the inhibitor (Supplementary Figure 4e).

Next, we evaluated whether GRK2 is required for the uncoating process, in which the M1 protein core dissociates from the vRNPs, allowing their release into the cytosol of the infected cells. A549 cells were infected with A/WSN/33 at an MOI = 25 PFU/cell for 3 h in the presence of the inhibitor or equivalent amounts of the solvent DMSO. To determine the efficiency of the uncoating process, we performed confocal immunofluorescence analysis with an M1-specific antibody, which targets an epitope that is solely accessible after uncoating[43], and quantified the intensity of the staining using ImageJ. In DMSO-treated cells, as a consequence of the uncoating of the virions, the M1 epitope recognized by the antibody is exposed and a green staining was

observed across the cytoplasm. However, upon BafA1 treatment or GRK2 inhibition, the intensity of the staining was significantly reduced and confined to discrete structures in the cytoplasm (Fig. 4f). Altogether, these results map the requirement of GRK2 to the viral uncoating process.

Given the role of the viral M1, M2 and NP proteins in uncoating, we also analyzed the phosphorylation status of these proteins in the absence or presence of a GRK2 inhibitor. We did not observe phosphorylation of M1 or NP, but we found that M2 was phosphorylated (Supplementary Figure 5), which is in line with published results on the phosphorylation status of M2 in virus particles[44]. However, no difference was observed between inhibitor-treated and untreated samples (Supplementary Figure 5) suggesting that GRK2 exerts its proviral function via a cellular rather than a viral target.

**GRK2 inhibition limits IAV in human airway cultures and mice.** Next, we tested the antiviral activity of paroxetine, a selective serotonin reuptake inhibitor (SSRI) recently shown to also inhibit GRK2 activity in vitro, in cell culture and in mice[45,46] (Fig. 5a). Paroxetine efficiently blocked IAV replication in a dose-dependent manner, being slightly more efficient than the GRK2 inhibitor used in our previous experiments. The control SSRI fluoxetine did not inhibit IAV infection (Supplementary Figure 6a) suggesting that paroxetine inhibited IAV via GRK2 inhibition rather than its function as SSRI. While A549, MRC5 and Wi38 are widely used models for IAV infection, they do not represent the complexity of the human airway epithelia. We therefore evaluated the antiviral activity of both GRK2 inhibitors in primary human bronchial epithelial cells differentiated into pseudostratified human airway epithelial (HAE) cultures under "air-liquid-interface" (ALI) conditions[47]. Fully differentiated HAE cultures from different donors were pre-treated with inhibitors for 1 h and subsequently infected with human IAV strains A/Netherlands/602/2009 (pdmH1N1), A/Brisbane/59/07 (H1N1), A/HongKong/68 (H3N2) or A/Brisbane/10/07 (H3N2) for 4 h in the presence of the inhibitors (Fig. 5b). Both compounds blocked nuclear accumulation of NP efficiently for all virus strains, with paroxetine being more potent. We then infected HAE cultures with the different strains of IAV and collected supernatants 16 and 24 hpi to quantify virus production by plaque assay (Fig. 5c–f). Paroxetine reduced virus titers by 100 to 1000-fold for all strains tested, confirming that blocking GRK2 activity could be used to limit IAV infection in human lung epithelia. Finally, we tested whether chemical inhibition of GRK2 could reduce viral replication and severity of infection in mice. As no data were available for the stability and bioavailability of GRKi in vivo, we selected paroxetine, which had been used in mice before, for our experiments[46]. 9 week old C57BL/6 J mice were injected intraperitoneally (i.p.) with 5 mg/kg paroxetine or an equivalent amount of solvent. At 24 h post treatment, mice were challenged with 10 PFU of a mouse-adapted A/Netherlands/602/2009 (pdmH1N1) or inoculated with phosphate-buffered saline (PBS) via the intranasal route. Paroxetine or solvent treatment was repeated on the day of infection and on days 1 (d1) and d2 post infection (pi). For one group of mice, weight loss and survival were monitored but no significant changes were observed between the paroxetine-treated group and the solvent control group (Supplementary Figure 6b, c). For a second group of mice, virus titers on d2 and d4 pi were determined from snout and lung homogenates to track the progression of the infection in the upper (URT) and lower respiratory tract (LRT) (Fig. 5g, h). No significant differences were observed in virus titers from d2 pi. However, paroxetine treatment decreased virus titers significantly in both organs on d4 pi. In summary, our results show that

inhibition of GRK2 led to strongly reduced viral titers in primary human airway cultures and also limited viral replication in a mouse model. However, GRK2 inhibition could not protect mice from weight loss or death in a lethal challenge model. While further in vivo studies using more potent GRK2 inhibitors will be needed to fully assess the potential of GRK2 inhibition as therapeutic option, our results establish GRK2 as novel drug target candidate for influenza.

## Discussion

Binding of IAV to its host cell triggers complex signaling cascades that regulate internalization of the virus but can also impact later events in the infection cycle. In order to obtain a global picture of the virus-induced signaling events in an unbiased manner, we applied quantitative phosphoproteomics to cells infected with influenza viruses for 5 or 15 min and revealed the unique IAV-induced phosphorylation signature. Even though our approach was limited to the identification of phospho-serine and -threonine containing peptides and did not include phospho-tyrosines, we uncovered >200 infection-induced changes in the phosphorylation status of cellular proteins within minutes of virus addition. We found that the MAPK pathway was the most prominent signaling cascade induced. While it was known before that this pathway becomes activated in response to IAV[6,48], our study reveals its central role in early stages of influenza virus infection.

As our main goal was to uncover novel drug targets for influenza, we focused on the identification of kinases that could be responsible for the observed changes in phosphorylation. A similar analysis for phosphatases would be equally interesting but thus far, bioinformatic tools for phosphatase prediction are less well developed. We found GRK2, ULK3, ROCK1 and PAK1 predicted to be involved in phosphorylation events identified at both time points analyzed. While a role for ROCK1 and PAK1 in IAV entry has been proposed before[12,13,30], GRK2 and ULK3 have not been implicated in influenza virus infections yet. ULK3 can interact with the ESCRTIII complex and functions in cytokinetic abscission. Specifically, ULK3 is required for delaying cell division if the chromosomes have not segregated yet[49]. With regard to viruses, ULK3 was found to be upregulated in cytomegalovirus infection but the functional relevance of this observation is unclear[50]. To our knowledge, no specific inhibitors of ULK3 have been developed yet and thus, the suitability of ULK3 as potential drug target for influenza is currently limited. In contrast, GRK2 is the target of intense research efforts. This kinase is best known for its role in G-protein-coupled receptor (GPCR) signaling, where GRK2 is recruited to the receptor upon activation, phosphorylates the GPCR, which in turn leads to binding of beta-arrestin to the receptor. This impairs downstream signaling and thereby leads to rapid desensitization. Furthermore, beta-arrestin binding also triggers receptor internalization and subsequent lysosomal degradation[51]. Increased levels of GRK2 are therefore associated with a reduction in GPCR signaling and this is particularly well documented for beta-adrenergic receptor signaling in heart tissue, where increased levels of GRK2 are linked with impaired cardiac contractility[51,52]. Hence, GRK2 has emerged as promising drug target in heart disease and inhibitor development is under way[53,54], which could eventually enable drug repurposing for the treatment of influenza virus infections.

At this point, the target(s) of GRK2 through which the kinase exerts its proviral function for IAV are not known. As described above, beta-arrestin is the best-studied substrate of GRK2 but we did not detect any phosphopeptides derived from beta-arrestin in our proteomic analysis. Furthermore, data from a meta-analysis of several genome-wide RNAi screens for required host factors of

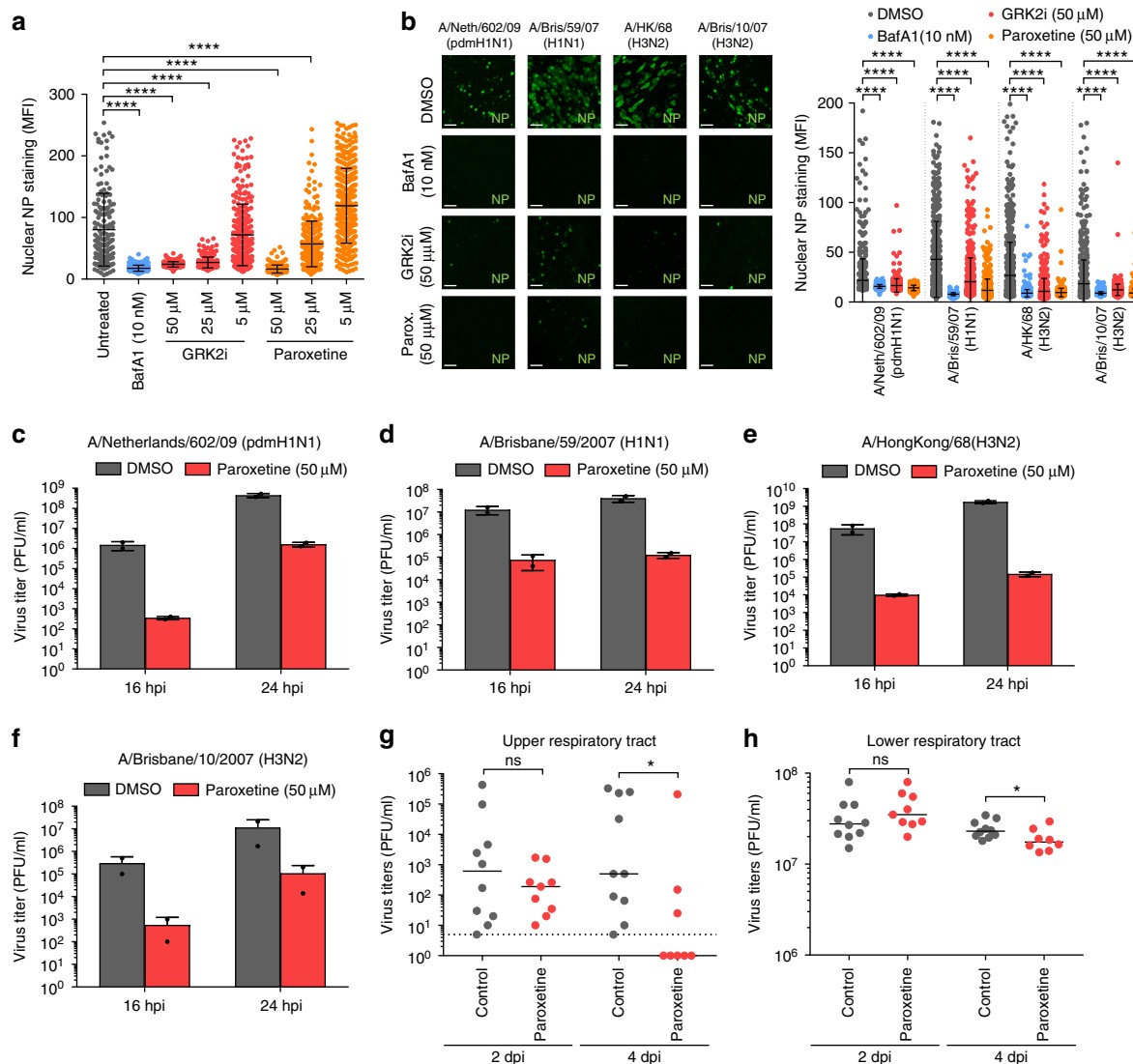

**Fig. 5** IAV replication can be limited by the use of GRK2 inhibitors in human airway cultures and in mice. **a** A549 cells were treated for 1 h with the indicated amounts of the different compounds and infected on ice with IAV A/WSN/33 (H1N1) at an MOI = 5 PFU/cell. Cells were incubated at 37 °C for 3 h in presence of the inhibitors, fixed and stained for viral nucleoprotein (NP, green) and nuclei (blue). The distribution of NP at the different time-points was analyzed by confocal microscopy and mean fluorescence intensity (MFI) in the cell nuclei (n > 200) was quantified using ImageJ. One representative out of four independent experiments is shown. Statistical significance was determined by unpaired t-test. **b** Fully differentiated human airway epithelial (HAE) cultures were infected with the indicated human IAV strains ($10^7$ PFU/well) in presence of the indicated compounds. Cells were processed and NP nuclear staining was quantified as described in (**a**). Scale bar corresponds to 40 μm. One representative out of at least two independent experiments is shown. **c–f** HAE cultures were infected with the indicated human IAV strains (1000 PFU/well) in presence or absence of paroxetine. At the indicated times post-infection, supernatants were harvested and virus titers were determined by plaque assay. For **a–f**, error bars represent standard deviation and statistical significance was determined by unpaired t-test. **g**, **h** Virus titers in the upper respiratory tract (URT) and lower respiratory tract (LRT) of mice treated with solvent (gray) or paroxetine (red) and infected with A/Netherlands/602/2009 (pdmH1N1) for 2 and 4 d pi are depicted (two independent experiments, n ≥ 8 per group). Statistical significance was determined using non-parametric two-tailed Mann–Whitney test. For all panels, ns (non-significant) = P > 0.05; *P ≤ 0.05; **P ≤ 0.01; ***P ≤ 0.001; ****P ≤ 0.0001

IAV[48] suggest that beta-arrestin is not a required host factor for IAV, and therefore most likely does not mediate the proviral function of GRK2. Of note, GRK2 was recently described as key stimulator of histone deacetylase 6 (HDAC6)[36], which in turn has been proposed to be recruited to viral fusion sites and enable efficient uncoating in a complex mechanism involving the aggresome processing machinery[43]. However, GRK2-mediated phosphorylation of HDAC6 has been shown to stimulate its α-tubulin deacetylase activity, which is not required for viral uncoating. In addition, no HDAC6-derived phosphopeptide was detected in our analysis suggesting that GRK2 does not exert its

proviral role through HDAC6. As we also did not observe any changes in the phosphorylation status of the viral proteins M1, M2 and NP in the presence of a GRK2 inhibitor, we speculate that GRK2 supports the viral uncoating process by phosphorylating non-canonical cellular targets. Further characterization of the interplay between GRK2, the components of the aggresome system and the cellular cytoskeleton will help to reveal the molecular mechanism of IAV uncoating.

Interestingly, GRK2 was recently shown to promote both, entry and RNA synthesis, of yellow fever virus (YFV), dengue virus (DENV) and Hepatitis C virus (HCV)[55], which are flaviviruses

causing severe disease in humans. Together with our findings that GRK2 is a key kinase involved in the initial steps of IAV infection, this suggests that inhibition of GRK2 function may represent a promising opportunity for the development of broad spectrum antivirals. In summary, our work reveals the unique phosphorylation signature induced by IAV within minutes of infection and identifies GRK2 as lead target for host cell-directed antivirals.

## Methods

**Chemicals and reagents**. The following compounds were used: dimethyl sulfoxide (DMSO, Sigma-Aldrich), bafilomycin A1 (Sigma-Aldrich), AT7519 (CDK inhibitor, Selleckchem), ulixertinib (MAPK inhibitor, Selleckchem), IPA-3 (PAK1 inhibitor, Selleckchem), Y-27632 (ROCK1 inhibitor, Selleckchem), methyl 5-[2-(5-nitro-2-furyl)vinyl]-2-furoate (GRK2 inhibitor, Santa Cruz Biotechnology), cyclo-heximide (Sigma-Aldrich) and paroxetine (GRK2 inhibitor, Sigma-Aldrich). For western blotting and immunofluorescence, the following primary antibodies were used: rabbit polyclonal anti-GRK2 antibody (C-15, Santa Cruz Biotechnology, sc-562, western blot 1:1000 dilution), mouse monoclonal anti-β-actin antibody (C4, Santa Cruz Biotechnology, sc-47778, western blot 1:1000 dilution), rabbit polyclonal anti-hEGFR (1005, Santa Cruz Biotechnology, sc-03-G, immunofluorescence 1:100 dilution, western blot 1:1000 dilution), mouse monoclonal anti HA-Tag (Cell Signaling #2367, immunofluorescence 1:100 dilution), rabbit polyclonal anti-NP antibody for western blotting (kind gift of A. Nieto, Centro Nacional de Biotecnología, Madrid, Spain, western blot 1:1000 dilution), rabbit polyclonal anti-NP antibody for immunofluorescence (kind gift of P. Palese, Mount Sinai Hospital, NY, USA, immunofluorescence 1:2000 dilution), mouse monoclonal anti-NP (ATCC HB65, H16-L10-4R5, immunofluorescence 1:5 dilution), mouse monoclonal anti-M1 antibody (ATCC HB64, M2-1C6-4R3, immunofluorescence 1:5, western blot 1:5 dilution), rabbit polyclonal anti-IAV M2 (ThermoFisher, PA5-32233, western blot 1:1000 dilution) and mouse monoclonal anti-EEA1 antibody (BD Bioscience, #610457, immunofluorescence 1:2000).

**Cells and viruses**. A549 (CCL-185), MRC5 (CCL-171) and Wi38 (CCL-75) were obtained from ATCC. Hep-2 were a kind gift of J. Pavlovic, Institute of Medical Virology, Zurich, Switzerland and MDCK were kindly provided by Ben Hale, Institute of Medical Virology, Zurich, Switzerland. A549, Hep-2 and MDCK cells were maintained in Dulbecco's modified Eagle medium (DMEM) supplemented with 10% (v/v) fetal bovine serum (FBS) and 1% (v/v) penicillin and streptomycin (Life Technologies). MRC5 and Wi38 were maintained in minimal essential Eagle medium (MEM, Sigma-Aldrich) supplemented with 10% (v/v) fetal bovine serum (FBS) and 1% (v/v) penicillin and streptomycin (Life Technologies). Isolation and cultivation of primary human bronchial epithelial cells to form pseudostratified/differentiated human airway epithelial (HAE) cultures was performed as described previously[56,57]. Influenza virus strains A/WSN/1933 (H1N1), A/Udorn/72 (H3N2), A/Netherlands/602/2009 (pdmH1N1), A/Brisbane/59/07 (H1N1), A/HongKong/68 (H3N2) or A/Brisbane/10/07 (H3N2) were kindly provided by P. Palese, Mount Sinai Hospital, NY, USA. Strain A/FPV/Dobson/34 (H7N7) was a kind gift of J. Pavlovic, Institute of Medical Virology, Zurich, Switzerland. All virus stocks were grown in 10-day old embryonated chicken eggs and titered by plaque assay on MDCK cells.

**Virus infection and titration**. For virus infection, cells were washed once with PBS supplemented with 20 μM Mg$^{2+}$, 10 μM Ca$^{2+}$, 0.3% (w/v) bovine serum albumin (BSA), and 1 % (v/v) penicillin-streptomycin (infection PBS). In case of HAE cultures, three washes with infection PBS were performed. MOI-adjusted inoculum size was subsequently added to the cells, which were incubated for 1 h at room temperature (RT) or for 1 h on ice, in case infection synchronization was needed. After the adsorption time, the inoculum was removed and replaced by an appropriate amount of post-infection medium (DMEM supplemented with 0.3% BSA, 20 mM HEPES, and 1% penicillin-streptomycin) and cells were incubated for the indicated times at 37 °C. Infectious particles in the supernatants of the cells were quantified by plaque assay on MDCK cells.

**Receptor Tyrosine kinase (RTK) profiling**. For the initial analysis included in Supplementary Figure 1a, a Proteome Profiler Human Phospho-RTK Array Kit (R&D Systems) was used. Briefly, A549 cells were seeded in 6-well plates and serum-starved overnight. The following day, cells were infected with A/WSN/33 at an MOI = 25 PFU/cell for 5 or 15 min, lysed and 300 μg of each lysate were used according to manufacturer's protocol. The membranes were read using a Luminescent Image Analyzer LAS-4000 Systems (FujiFilm) and signal intensities were quantified by ImageJ.

**SILAC labeling and preparation of cell extracts**. Five biological replicates of the phosphoproteomic experiment were performed using SILAC-labeled A549 cells. For each replicate, two T150 flasks of cells labeled with each of the following conditions were used: L (light), where A549 cells were grown in normal DMEM; M (medium), where A549 cells were grown in DMEM containing 4,4,5,5-D$_4$ lysine

and $^{13}C_6$ arginine; and H (heavy), where A549 cells were grown in DMEM containing $^{13}C_6$ $^{15}N_2$ lysine and $^{13}C_6$ $^{15}N_4$ arginine. Incorporation of the different isotopes was confirmed by LC-MS/MS. Medium and heavy labeled cells were infected with IAV (A/WSN/1933) at an MOI = 25 PFU/cell for 5 and 15 min, respectively. Virus inoculum was prepared in infection PBS. As a mock-infection control, an equivalent amount of allantoic fluid from non-infected 10-day old embryonated chicken eggs was diluted in infection PBS and added to unlabeled cells (L). After the corresponding incubation time, the cells were directly lysed in SDS-lysis buffer (4% (w/v) SDS, 100 mM Tris/HCl pH 8.2, 0.1 M DTT, Complete (Roche) and Phos-Stop (Roche)) and the different cell extracts were pooled and sonicated briefly to reduce viscosity. In order to concentrate the samples, proteins were precipitated using four volumes of −20 °C acetone and incubated overnight at −20 °C. Following centrifugation (11,000 × $g$, 1 h, 4 °C), the precipitated protein pellet was washed three times with −20 °C acetone, and then dried for 30 min. Protein pellets were solubilized in a smaller volume of SDS-lysis buffer and the protein concentration of the different replicates was determined using Qubit fluorometer (Invitrogen).

**Trypsin digestion and phosphopeptide enrichment**. Protein extracts were processed using filter-aided sample preparation (FASP)[58]. Briefly, 500 μg of protein were diluted in urea buffer (8 M urea in 100 mM Tris/HCl pH 8.2) and loaded into Microcon-30kDa Centrifugal Filter Units with Ultracel-30 membranes (Millipore). The proteins were bound to the membrane by centrifugation and alkylated using a 50 mM iodoacetamide (IAA) solution in urea buffer. After washing three times with urea buffer, the proteins were washed twice with 500 mM NaCl and three times using a 50 mM triethylammonium bicarbonate (TEAB) buffer pH 8.5. Overnight on-filter digestion was performed with trypsin (Promega) at a ratio of 1:50 (w/w) trypsin to protein, in a wet chamber at 37 °C. The tryptic peptides were eluted from the filter by centrifugation and a second elution using 50 mM TEAB buffer was performed to increase the yield. A fraction of these peptides was collected for total proteome analysis. Phosphopeptides were enriched by Ti$^{4+}$-immobilized metal affinity chromatography (Ti-IMAC) and TiO$_2$ affinity chromatography using magnetic microspheres (ReSyn Biosciences) and a magnetic separator (Thermo Fisher Scientific). 1 mg of Ti-IMAC or TiO$_2$ magnetic microspheres were washed twice with 70% (v/v) ethanol and equilibrated once in 1% NH$_4$OH and three times in phosphopeptide loading buffer (1 M glycolic acid in 80% (v/v) acetonitrile (ACN), 5% (v/v) trifluoroacetic acid (TFA)). 500 μg of tryptic peptides were diluted in loading buffer, added to 1 mg of equilibrated magnetic microspheres and incubated for 20 min at RT with continuous rotation. After binding of the phosphopeptides, the microspheres were washed once with loading buffer and unbound peptides were removed by three washes with washing buffer (80% (v/v) ACN, 1% (v/v) TFA). Phosphopeptides were eluted by incubation of the microspheres for 15 min in elution buffer (1% (v/v) NH$_4$OH), acidified by addition of 10% (v/v) formic acid (FA) and dried in a SpeedVac concentrator (Thermo Fisher Scientific). Peptides were resuspended in 3% (v/v) ACN, 0.1% (v/v) TFA and desalted using C18 ZipTips (Millipore). Filters were equilibrated once in a 60% (v/v) ACN, 0.1% (v/v) TFA and subsequently twice in 0.1% (v/v) TFA solution. Peptides were bound to the filters, washed three times with 0.1% (v/v) TFA and eluted in 60% (v/v) ACN, 0.1% (v/v) TFA. After drying them in a SpeedVac concentrator, peptides were dissolved in 10 μl of LC-MS/MS solvent (3% (v/v) ACN, 0.1% (v/v) FA) prior to LC-MS/MS analysis.

**LC-MS/MS and initial data analysis**. All data were acquired on an Orbitrap Fusion Tribrid mass spectrometer (Thermo Scientific, San Jose, Ca), which was connected to an Easy-nLC 1000 HPLC system (Thermo Scientific). 4 μl of the peptide samples in 0.1% formic acid were loaded onto a frit column (inner diameter 75 μm, length 15 cm) packed with reverse phase material (C18-AQ, particle size 1.9 μm, pore size 120 Å, Dr. Maisch GmbH, Germany), and separated at a flow rate of 300 nl/per min. Solvent composition of buffer A was 0.1% formic acid in water, and buffer B contained 0.1% formic acid in acetonitrile. For the phospho-enriched samples, the following LC gradient was applied: 0 min: 2% buffer B, 110 min: 30% B, 120 min: 50% B, 122 min: 95% B. Survey scans were recorded in the Orbitrap mass analyzer in the range of m/z 350–1500, with a resolution of 60,000 and a maximum injection time of 50 ms. Higher energy collisional dissociation (HCD) spectra were acquired in the ion trap mass analyzer, using a normalized collision energy of 28%. The precursor ion isolation width was set to m/z 2.0, and a maximum injection time of 100 ms and an AGC target value of 3e$^4$ were applied. Charge state screening was enabled, and charge states 2–7 were included. The threshold for signal intensities was 1e$^4$, and precursor masses already selected for MS/MS acquisition were excluded for further selection during 30 s. For total protein samples, the following LC gradient was applied: 0 min: 5% buffer B, 50 min: 25% B, 60 min: 32% B, 70 min: 97% B. Survey scans were recorded in the range of m/z 300–1500, with a resolution of 120,000 and a maximum injection time of 50 ms. HCD spectra were acquired in the ion trap, using a normalized collision energy of 28%, an isolation window of m/z 1.6, a maximum injection time of 250 ms and an AGC target value of 1e$^2$. The intensity threshold was 5e$^3$ and the dynamic exclusion duration was 25 s. A first round of peptide identification was conducted in Mascot (www.matrixscience.com)[59] and the phosphoenrichment efficiency and the number of identifications for the different samples was evaluated using Scaffold PTM 3.0. The mass spectrometry proteomics data have been

deposited in the ProteomeXchange Consortium via the PRIDE[60] partner repository with the data set identifier PXD009999.

**MaxQuant analysis.** Data acquired on the Orbitrap Fusion were analyzed with MaxQuant v1.5.0.30[23], searching a target-decoy database consisting of human (88,708 forward Uniprot entries) and influenza A virus (A/WSN/33, 9 forward Uniprot entries) sequences. As an additional control, a database containing chicken sequences (43,340 forward Uniprot entries) was used for the identification of IAV-induced chicken proteins in our egg-grown stock. To identify contaminant proteins, the internal contaminant database of MaxQuant was used. Search parameters were: Cysteine carbamidomethylation as fixed modification, protein N-terminal acetylation and methionine oxidation and phosphorylation of serine, threonine and tyrosine as variable modifications; SILAC labeling (Arg10, Lys8, Arg6, Lys4) as heavy labels; enzyme trypsin/P; two missed cleavages were allowed; and a minimum of seven amino acids per identified peptide were required. The precursor ion mass tolerance was set to 7 ppm, and the fragment mass tolerance was set to 20 p.p.m. Peptide identifications were accepted until less than 1% of the reverse hits were retained in the list. The protein false discovery rate (FDR) was set to 1%. For each phosphopeptide in the data set, the median of the MaxQuant ratios for the different replicates was calculated and the data from both TiO$_2$ and Ti-IMAC based approaches were combined for further analysis.

**Pathway enrichment analysis.** To identify the top enriched pathways, the distributions of annotated KEGG pathways[22] among differentially phosphorylated peptides at 5 and 15 min post infection (log$_2$FC ≥ 0.5) were compared against the background of all detected proteins in our MS analysis plus the ones previously identified in a phosphoproteomic characterization of HIV infection[61] (Supplementary Figure 7). KEGG pathways for the whole population were retrieved (R packages org.Hs.eg.db, v. 3.2.3, KEGG.db, v. 3.2.2) and a standard hypergeometric test was performed followed by Bonferroni correction. Enrichment results are presented in the context of the KEGG pathway topology using igraph (v.1.0.1).

**Kinase prediction.** The kinase prediction is derived from high stringency scores calculated using the Group-based Prediction System 3.0 (GPS 3.0, http://gps.biocuckoo.org/online.php)[27–29], which ranks the likelihood that a particular kinase or kinase family phosphorylates a given residue considering the amino acids surrounding the phosphorylation site. All predictions for the experimentally detected phosphorylation sites were ranked based on the difference between the site threshold and the site prediction score (Supplementary Figure 7). In order to determine the selection parameters we would use, given that more than one kinase may be responsible for the phosphorylation of a given protein at a given position, we first worked on a subset of entries from Phospho.ELM (v.9.0, April 2015, http://phospho.elm.eu.org/dataset.html). Briefly, we downloaded all abstracts ($n = 1690$) associated with all kinase-phosphosite pairs (rentrez 1.0.4) and selected those reported in the context of viral infection ($n = 160$ unique sites on 51 unique proteins, representing substrates of 12 major kinase groups, Supplementary Figure 2a). We subsequently ran GPS 3.0 on the full sequences of these proteins and checked for conditions that allowed us to retrieve as many true positives per phosphosite $Pi$ from all predictions for $Pi$ (Supplementary Figure 2b, c) using the most stringent (2% false positive rate for serine/threonine kinases and 4% for tyrosine kinases) prediction threshold. The top five predictions in terms of the GPS-calculated score, which indicates the similarity of a detected phosphosite to a phosphosite in the gold standard Phospho.ELM; or the derived parameter Delta, defined as the difference between the GPS-calculated score and the GPS-provided cutoff per position, were found to include 70% of the true positives in this scenario and was chosen as the cutoff for the kinase-phosphosite pairs that we report. Top predicted kinases for the differentially phosphorylated sites at both 5 and 15 min post infection (log$_2$FC ≥ 0.5) were considered as IAV-responsive kinases. The distribution of the top predicted kinases was then compared against the background of the total detected peptide population and enriched hits were identified. To determine if the enrichment for a particular kinase is significant, we performed a 1000-fold random selection of peptides from all peptides detected in our assay, where the number of peptides sampled per randomization is equal to the size of the hyperphosphorylated peptides at 5 or 15 min post infection (log$_2$FC > 0.5 wrt mock). We subsequently compared the frequencies at which a top kinase is predicted for detected versus randomized conditions using Fisher's exact test. For a subset of the top predicted kinases (CDK1, CDK5, GRK2 and MAP2K1), significant enrichment cannot be established, given that their higher and more variable scores ($\mu_{CDK5,GRK2,CDK1,MAP2K1} = 17.34 \pm 5.33$, $\mu_{other} = 7.03 \pm 1.33$) ensure their inclusion in the top ranks for any phosphosite for which these kinases are predicted (Supplementary Figure 2c). Others, including PAK1 and CBK1, tend towards enrichment ($p$ value < 0.1 (Fisher's exact test)) and, consequently, we cannot exclude their potential contribution to viral replication and included them in our analysis. The computer code is available upon request.

**Phosphorylation site consensus motifs.** To find motifs potentially linked to phosphorylated sites associated with a kinase prediction, we first grouped sequences within +7 to +1 amino acids of the detected phosphorylation site (i.e., the phosphorylation site is at position 0); sequences can be included in more than

one group based on the assumption that different kinases may be responsible for phosphorylation of a given protein at a given position. Sequences linked to each predicted kinase group were submitted to WebLogo (http://weblogo.berkeley.edu/logo.cgi) for motif detection (Supplementary Figure 7).

**Inhibitor testing with WSN-Ren reporter virus.** In order to quantitatively measure the effect of the different chemicals on viral replication, we used a reporter IAV derived from A/WSN/33, in which the coding sequence of the hemagglutinin protein was replaced by that of Renilla luciferase to allow real-time determination of viral replication[62]. The different kinase inhibitors were added to the A549 cells together with the reporter virus at an MOI = 1 PFU/cell. After synchronizing infection by allowing the virus to bind to target cells for 1 h at 4 °C, the inoculum was removed and fresh culture medium containing the Renilla luciferase substrate (EnduRen Live Cell Substrate, Promega) and the indicated concentrations of the different inhibitors were added to the cells. The plates were transferred to 37 °C and real-time luminescence measurements were performed at the different time-points indicated using an EnVision Multilabel Reader (Perkin Elmer).

**siRNA transfection and cell viability assay.** A549, MRC5 or Wi38 cells were reverse transfected with 30 nM siRNA (Qiagen) diluted in Opti-MEM (Life Technologies) using the RNAiMAX reagent according to the manufacturer's protocol (Invitrogen). At 48 h post-transfection, cells were either infected or cell viability was determined using the CellTiter-Glo assay (Promega). In addition, the CellTiter-Glo kit was used to assess the effect of the different inhibitors on A549 cells in the absence of infection.

**Western blot.** Cell extracts were prepared using Laemmli buffer (62.5 mM Tris-HCl pH 6.8, 25% glycerol, 2% SDS, 350 mM DTT, 0.01% Bromophenol Blue). Samples were subjected to standard SDS-PAGE and proteins were transferred to nitrocellulose membranes (Hybond ECL, GE Healthcare). Indicated primary antibodies were used with fluorescent secondary antibodies from Li-Cor and images were acquired on a Odissey Fc imaging system.

**Zn$^{2+}$-Phos-tag SDS PAGE gels.** For sample preparation, cells were lysed using a pH neutral buffer (20 mM TrisHCl pH 8.1, 0.5% (v/v) NP-40, 200 mM NaCl, cOmplete EDTA-free protease inhibitor cocktail (Roche) and PhosSTOP (Roche)) before Laemmli buffer was added. Gels were cast according to manufacturer's protocol (wako-chem) using Phos-tag™ AAL-107 (MW: 595, Wako Cat. No. 304-93525). Samples were run on the Zn$^{2+}$-Phos-tag SDS PAGE gels for 4 h at 60 V and were blotted over night at 4 °C onto nitrocellulose membranes (Hybond ECL, GE healthcare). Staining and analysis were done as described for standard SDS-PAGE. For the blots shown in Fig. 3d, e, the uncropped versions of the blots are provided in Supplementary Figure 8.

**GRK2 translocation assay.** Hep-2 cells were grown on glass coverslips in 24-well plates and transfected with HA-GRK2 (a kind gift from Dr. U. Quitterer, ETH Zurich, Switzerland) and pcDNA6A-myc-EGFR (a kind gift from Mien-Chie Hung (Addgene plasmid # 42665[63])) using ViaFect (Promega) as transfection reagent. After infection with A/WSN/1933 with an MOI = 100 PFU/cell or stimulation with hrEGF (100 ng/ml), cells were processed for immunofluorescence as described below.

**Immunofluorescence staining and confocal microscopy.** For immunofluorescence staining, A549, Hep-2 or MDCK cells were grown on glass coverslips in 24-well plates. In the case of HAE cultures, cells were fixed and stained directly on the membrane. After infection cells were fixed with 3% paraformaldehyde (PFA) and permeabilized using immunofluorescence (IF) buffer (PBS supplemented with 50 mM NH$_4$Cl, 0.1% saponin, and 2% BSA). After 30 min in IF buffer, the cells were incubated overnight at 4 °C with the corresponding primary antibodies diluted in IF buffer. Cells were washed three times with IF buffer and incubated with the secondary antibodies diluted in DAPI-containing IF buffer for 1 h at RT. Alexa Fluor-conjugated donkey anti-mouse/rabbit antibodies (catalog no. A-21202, A-21206, A10036, and A10040; Life Technologies) were used as secondary antibodies. Cells were washed three times with IF buffer, before the coverslips or membranes were dipped in deionized water and inversely mounted onto glass microscope slides using ProLong Gold Antifade Mountant (Thermo Fisher Scientific). All microscopy images were acquired with a confocal laser scanning microscope (Leica SP5). The quantification of the nuclear NP signal was performed using an automated macro in ImageJ. Statistical analysis of the different images was done using PRISM software. For the endosomal co-localization analysis, Imaris software was used with standard co-localization settings and masks.

**Virus-like particle (BlaM1 VLP) production and infection.** A/WSN/33-based VLPs harboring β-lactamase-M1 (BlaM1) fusion proteins were produced essentially as described by Tscherne & García-Sastre[42]. Briefly, HEK 293 T cells seeded onto poly-L-lysine-coated (Sigma-Aldrich) 6-well plates were transfected in Opti-MEM (Life Technologies) with 2 µg BlaM1, 500 ng pCAGGS-WSN-HA, 500 ng

pCAGGS-WSN-NA per well using ViaFect (Promega) as the transfection reagent (3 μl ViaFect/μg DNA). Medium was exchanged 6 h post-transfection with Opti-MEM (Life Technologies) containing 1% penicillin-streptomycin. VLPs were collected 72 h after transfection and treated with 5 μg/mL TPCK-trypsin (Sigma-Aldrich) for efficient HA cleavage. Trypsin was inactivated with 10 μg/mL trypsin inhibitor from Glycine max (Sigma-Aldrich). Prior to infection, MDCK cells were treated with the indicated concentrations of the GRK2 inhibitor for 1 h. For infection of MDCK cells in 24-well plates, 200 μl BlaM1 VLPs were added to the cells in the presence of the compound and incubated for 4 h at 37 °C. Cells were harvested by trypsinization and incubated with the florigenic substrate CCF2-AM (ThermoFisher Scientific). Cells were analyzed on a FACSVerse System (BD) and dead cells excluded by a live/dead staining (LIVE/DEAD Fixable Near-IR Dead Cell Stain Kit, ThermoFisher Scientific).

**Mouse experiments**. 9 week old C57BL/6J mice (Charles River, France) were injected i.p. with 5 mg/kg paroxetine in 5% DMSO in PBS or 5% DMSO in PBS (solvent) 24 h prior to infection. Mice were infected after ketamine (100 mg/kg)/ xylazine (5 mg/kg) anaesthesia (i.p.) with 10 PFU of A/Netherlands/602/2009 (pdmH1N1) (kindly provided by Dr. F. Krammer, Icahn School of Medicine at Mount Sinai, New York) or inoculated with PBS via the intranasal route. Paroxetine or solvent treatment was repeated on d0, d1 and d2 pi. The first group of mice was sacrificed on d2 and d4 pi by controlled $CO_2$ exposure and lungs and snout were removed aseptically. Organs were homogenized in 1 ml sterile PBS using sterile 1/4" steel or ceramic beads in a 2 ml screw cap tube using a BeadBlaster 24 Position Homogenizer (Benchmark Scientific) at 6 m/s for two times 30 s with a 30 s break on ice. Homogenates were precipitated at 10,000 g for 10 min and clear supernatants were analyzed for infectious virus particles using standard plaque assay on MDCK. The second group of mice was monitored daily for weight loss and survival up to d7 when all infected mice reached humane endpoints. All animal procedures were in accordance with federal regulations (Schweizer Bundesamt für Veterinärwesen BVET), controlled by institutional (Direction de l'expérimentation animale d'universite de Geneve) and cantonal authorities (Commission cantonale de Geneve pour les expériences sur les animaux) and approved under license no GE/159/17.

## Data Availability
The mass spectrometry proteomics data have been deposited in the ProteomeXchange Consortium via the PRIDE60 partner repository [http://www.ebi.ac.uk/pride] with the data set identifier PXD009999. The authors declare that all other data supporting the findings of this study are available within the article and its Supplementary Information files, or are available from the authors upon request.

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

## Acknowledgements

This work was supported by the Swiss National Science Foundation (grant 31003A_156805 to SST) and by the Swiss Initiative in Systems Biology, SystemsX, through a fellowship (2013/137) provided to M.P.D., who was hosted by M. Delorenzi of the Bioinformatics Core Facility, SIB Swiss Institute of Bioinformatics, Lausanne, Switzerland.

## Author Contribution

E.Y. designed and performed the experiments, analyzed the data, prepared the figures and wrote the paper. A.H. and S.Y. designed and performed experiments, analyzed the data, prepared figures and contributed to manuscript preparation. M.P.D. performed the data analysis, prepared figures and contributed to manuscript preparation. S.Sc., E.E. and U.K., performed the experiments, and contributed to manuscript preparation. P.G. provided technical support for the mass-spectrometry. J.G. provided support for the analysis of mass-spectrometry data. R.D. provided the primary human airway cultures. M.S. performed and supervised the animal work and contributed to manuscript preparation. S.St. planned and supervised the work, acquired funding, designed experiments and wrote the paper.

## Additional information

**Competing interests:** The authors declare no competing interests.

