## [Peer Review File · Nature Communications]

Reviewers' comments:

Reviewer #1 (Remarks to the Author):

In this manuscript Yángüez et al. used a combination of quantitative phosphoproteomics, kinase motif predication and in vitro and in vivo experiments to discover and validate potential kinase drug targets that would show promise for the new antiviral therapy for seasonal influenza. Global phosphoproteome screen was performed in A549 cells infected for 5 or 15 minutes with influenza A virus and the results were compared to those obtained from uninfected cells as a control. After the phosphoproteomics data analysis the authors focused on the GRK2 kinase and looked at the mechanism of its action and effects of its inhibition in cultured cells and in a mouse system. The experiments have been carefully performed and the paper is well written. Specific comments are outlined below.

1. Page 5, Lines 91-93: The authors state that their approach did not include phosphotyrosines. It is true that the specific phosphotyrosine IP has not been performed, however, as per the data from the supplemental table, 47 pY sites were identified, so perhaps the initial statement could be phrased differently.
2. Please deposit the raw phosphoproteomics data in the publicly accessible database (such as ProteomeXchange) as is the standard developed by the proteomics community and provide the accession number in the appropriate part of the Methods section.
3. The experiments on mice as described on page 13 (lines 296-304) included a control experiment where mice were treated with paroxetine but not infected with virus. However, even though this experiment obviously was performed, the effect of paroxetine alone is not shown in any of the figures and despite paroxetine being an approved inhibitor which no doubt has been tested on mice, the authors should provide the result of their treatment in comparison to the mice treated with solvent or untreated to check the viability and any other effects on their mouse strain in their experimental setting (inhibitor dose, housing conditions, or any strain modifications can affect the outcome of the treatment).
4. General comment to the figures with indicated statistical significance: please provide (in the figure legends) the p-values corresponding to the number of asterisks marked on each figure, as well as indicate that "ns" means "non-significant".

Reviewer #2 (Remarks to the Author):

Despite the increasing knowledge of influenza A virus (IAV), the early steps of IAV infection are not well elucidated. This study by Yanguéz et al. presents a first profiling of cellular serine/threonine phosphorylation events within 15 minutes of IAV infection. This work is important because elucidation of these mechanisms will ultimately help design a new type of antivirals to block viral infection before the virus starts replication in the nucleus. The authors further relied on bioinformatics to reveal several kinases that may involve the early phosphorylation events. One of these candidates, GRK2 was chosen for detailed study. The data for GRK2 is solid and convincing, however, the underlying mechanisms and in vivo efficacy need further clarification and stronger evidence.

Major points:

1. The in vivo study showed the decreased viral infection in the respiratory tract using 40 PFU, a low dose for mouse infection. However, the authors may examine mouse survival rate using a high dose of IAV in mock-treated vs. drug-treated mice. Lung histology examination may be also included to make the Figure 5.
2. The reviewer appreciates the effort made to define which step of early infection the GRK2 involves.

The authors concluded that GRK2 is critical for viral core uncoating from M1 proteins before entering cytoplasm. The current dogma is that the physical conditions, most importantly, the low pH environment inside the virion caused by the IAV proton channel protein M2, is essential for uncoating. Numerous in vitro experiments have agreed the indispensability of low pH condition, nevertheless, there should be other regulatory mechanisms inside the cells. The GRK2 regulation of uncoating is a novel discovery, however, the mechanism is not clear. First, GRK2 kinase activity is required, but the potential substrates involved in uncoating are not even discussed. Secondly, is GRK2 activated after IAV infection or is it constitutively activated? If GRK2 activation is induced by IAV, is there any phosphorylated peptide of GRK2 found in the phosphorylation profiling in the Figure 1? It is also interesting to know whether GRK2 phosphorylates viral proteins, such as M1, and the effects of such phosphorylation on uncoating process.

Minor points:

1. Line 85, to my understanding, "the phosphorylation" should be "the tyrosine phosphorylation" of EGFR. If it is true, what is the rationale not to include tyrosine phosphorylation profile?
2. Infection causes cell stress. Is there any cross-talk with the stress signaling pathway by network analysis?

Reviewer #3 (Remarks to the Author):

In this study a phosphoproteomic screen was performed with the aim to identify cellular kinases involved in influenza A virus infection. To this end, phosphoproteomic profiling was performed after 5 min and 15 min infection. Kinases activated in response to infection were identified via bioinformatics analysis. Inhibitors of several of the identified kinases were shown to inhibit infection. GRK2 was shown to be required for IAV uncoating. A SSRI that also inhibits GRK2 was shown to decrease IAV replication when administered prior to infection in mice. In general it is a very nice study that is well executed and written. However, there are a few issues, the authors should address.

1. It is not clear to me why the 5 and 15 min time points were chosen. The only reason seems to be the study that shows that at these time points EGFR phosphorylation was observed?
2. The moi used in the study mentioned in point 1 (ref 8) is very high. What is the moi (and viral strain) used in the phosphoproteomic screen in this study. How does this relate to the number of IAV particles per cells? How physiologically relevant are the moi used, and are similar phosphorylation profiles expected at lower moi?
3. Is it possible that part of the phosphorylation signature observed does not result from IAV particles attaching to cells, but from other factors induced and released from infected cells (e.g. interferons or interferon-stimulated gene products). The virus stocks do not seem to be purified. Would it be possible to perform control experiments using conditioned media from which particles are removed?
4. In the mouse experiment, the authors do not use the inhibitor that they used throughout the study, but rather a SSRI that also inhibits GRK2 activity. It cannot be excluded that the inhibition observed results from the drug acting as a SSRI rather than from inhibition of GRK2. Were animal studies also performed using the other GRK2 inhibitor and what was the outcome? The authors could perform inhibition experiment (at least in vitro) using another SSRI that not inhibits GRK2 (e.g. fluoxetine?) to show that SSRIs do not generally inhibit IAV infection and uncoating.

We would like to thank the reviewers for their fair and constructive criticism, which we have addressed point-by-point as outlined below. We believe that these experimental additions and changes to the text have improved the manuscript substantially. (The line numbers given below refer to the manuscript version with marked-up changes.)

Reviewer #1 (Remarks to the Author)

In this manuscript Yángüez et al. used a combination of quantitative phosphoproteomics, kinase motif predication and in vitro and in vivo experiments to discover and validate potential kinase drug targets that would show promise for the new antiviral therapy for seasonal influenza. Global phosphoproteome screen was performed in A549 cells infected for 5 or 15 minutes with influenza A virus and the results were compared to those obtained from uninfected cells as a control. After the phosphoproteomics data analysis the authors focused on the GRK2 kinase and looked at the mechanism of its action and effects of its inhibition in cultured cells and in a mouse system. The experiments have been carefully performed and the paper is well written. Specific comments are outlined below.

1. Page 5, Lines 91-93: The authors state that their approach did not include phosphotyrosines. It is true that the specific phosphotyrosine IP has not been performed, however, as per the data from the supplemental table, 47 pY sites were identified, so perhaps the initial statement could be phrased differently.

According to the reviewer's suggestion we rewored l. 96-97 and l. 110-11 to clarify that our approach did not enrich for phospho-tyrosines but nevertheless yielded quantitative results for a few phospho-tyrosine-containing peptides.

2. Please deposit the raw phosphoproteomics data in the publicly accessible database (such as ProteomeXchange) as is the standard developed by the proteomics community and provide the accession number in the appropriate part of the Methods section.

We have deposited the raw data in ProteomeXchange and added this information in l. 571-573. Upon publication of this manuscript, the data will be publicly available. Currently, the data are accessible for reviewers on the ProteomeXchange website (<http://www.ebi.ac.uk/pride>) using the following login details:

Username: reviewer67390@ebi.ac.uk

Password: zTzHyjJE

3. The experiments on mice as described on page 13 (lines 296-304) included a control experiment where mice were treated with paroxetine but not infected with virus. However, even though this experiment obviously was performed, the effect of paroxetine alone is not shown in any of the figures and despite paroxetine being an approved inhibitor which no doubt has been tested on mice, the authors should provide the result of their treatment in comparison to the mice treated with solvent or untreated to check the viability and any other effects on their mouse strain in their experimental setting

(inhibitor dose, housing conditions, or any strain modifications can affect the outcome of the treatment).

In response to the reviewer's comment we have included weight loss and survival data for inhibitor-treated uninfected mice in the new supplementary figures S7b-c, which show that the inhibitor-treatment alone did not affect weight loss or survival in the absence of infection. We also added this information to the text (l. 363-366).

4. General comment to the figures with indicated statistical significance: please provide (in the figure legends) the p-values corresponding to the number of asterisks marked on each figure, as well as indicate that "ns" means "non-significant".

We apologize for not including this information. In our revised version, this has been corrected and we have included the missing information in the figure legends.

Reviewer #2 (Remarks to the Author)

Despite the increasing knowledge of influenza A virus (IAV), the early steps of IAV infection are not well elucidated. This study by Yanguéz et al. presents a first profiling of cellular serine/threonine phosphorylation events within 15 minutes of IAV infection. This work is important because elucidation of these mechanisms will ultimately help design a new type of antivirals to block viral infection before the virus starts replication in the nucleus. The authors further relied on bioinformatics to reveal several kinases that may involve the early phosphorylation events. One of these candidates, GRK2 was chosen for detailed study. The data for GRK2 is solid and convincing, however, the underlying mechanisms and in vivo efficacy need further clarification and stronger evidence.

Major points:

1. The in vivo study showed the decreased viral infection in the respiratory tract using 40 PFU, a low dose for mouse infection. However, the authors may examine mouse survival rate using a high dose of IAV in mock-treated vs. drug-treated mice. Lung histology examination may be also included to make Figure 5.

In order to address this point we performed additional mouse experiments using paroxetine as GRK2 inhibitor and assessed weight loss and survival upon infection with influenza A virus. The new results are shown in supplementary figures S7b-c and the text has been updated accordingly (l. 363-366).

Unfortunately, we made a mistake in the previous version of our manuscript: We had listed 40 PFU as infection dose but our experiments had been done with 10 PFU. We apologize for this mistake and we corrected it in the revised version. 10 PFU of the mouse-adapted strain of pandemic H1N1 correspond to 5xLD50 under Swiss animal rights legislation, meaning that the dose used is lethal and already constitutes a high dose. We therefore used the same infection conditions as in our previous mouse experiments but measured weight loss and survival instead of virus titers. Given the modest (but significant) differences in lung virus titers it is probably not too surprising that no significant differences in weight loss or survival were detected. Furthermore, we also examined the lungs of mice

on d4 p.i. and compared mock-treated to inhibitor-treated animals (please see figure below). As expected, a clear difference can be seen between mock-infected and influenza A virus-infected animals but no difference was detectable between mock-treated and inhibitor-treated mice. These results show that GRK2 inhibition by paroxetine from d-1 up to d2 results in a significant reduction of virus titers upon challenge with a lethal dose of pandemic H1N1, but is not potent enough to protect from weight loss, death or lung pathology. We discuss these results in I. 371-377.

Fig. 1: 9 week old C57BL/6J mice were injected intraperitoneally (i.p.) with 5 mg/kg paroxetine or an equivalent amount of solvent. At 24 h post treatment, mice were challenged with 10 PFU of A/Netherlands/602/2009 (pdmH1N1) or inoculated with phosphate-buffered saline (PBS) via the intranasal route. Paroxetine or solvent treatment was repeated on the day of infection and on days 1 (d1) and d2 post infection (pi). On d4 mice were sacrificed and lungs were removed. Lungs were fixed in 4% formaldehyde, embedded in paraffin and cut. Slices were stained with hematoxylin and eosin (1). Representative images are shown.

2. The reviewer appreciates the effort made to define which step of early infection the GRK2 involves. The authors concluded that GRK2 is critical for viral core uncoating from M1 proteins before entering cytoplasm. The current dogma is that the physical conditions, most importantly, the low pH environment inside the virion caused by the IAV proton channel protein M2, is essential for uncoating. Numerous in vitro experiments have agreed the indispensability of low pH condition, nevertheless, there should be other regulatory mechanisms inside the cells. The GRK2 regulation of uncoating is a novel discovery, however, the mechanism is not clear.

First, GRK2 kinase activity is required, but the potential substrates involved in uncoating are not even discussed. Secondly, is GRK2 activated after IAV infection or is it constitutively activated? If GRK2 activation is induced by IAV, is there any phosphorylated peptide of GRK2 found in the phosphorylation profiling in the Figure 1? It is also interesting to know whether GRK2 phosphorylates viral proteins, such as M1, and the effects of such phosphorylation on uncoating process.

Activation of GRK2

We hypothesised that GRK2 is activated by IAV infection, as this would explain the GRK2 phosphorylation signature that we observed in infected samples. In order to test this experimentally, we assessed the phosphorylation status of GRK2 upon IAV infection. We had not detected phosphopeptides derived from GRK2 in our proteomic analysis and thus did not know which specific phosphorylation site to monitor. We therefore used the Phos-tag SDS-PAGE method (2), which allows for the visualization of differentially phosphorylated versions of a protein with standard antibodies. We tested and optimized our experimental approach by using EGF treatment as positive control and

show in the new figure 3d that we can detect a phosphorylated form of GRK2 upon EGF treatment. Using this system, we monitored GRK2 phosphorylation at early time points of IAV infection and observed a phosphorylated form of GRK2 appearing upon IAV infection. These new results are shown in figures 3d-e and described in l. 229-250.

In addition, we analysed whether GRK2 changes in localization upon IAV infection as it had been reported that activated GRK2 translocates to the plasma membrane. Indeed, we observed recruitment of GRK2 to the plasma membrane within minutes of IAV infection. These new results are shown in figure 3f and described in l. 251-262. Furthermore, we observed that the translocation of GRK2 was dependent on EGFR expression (figure 3g) suggesting that GRK2 is tyrosine-phosphorylated by EGFR and thereby activated upon IAV infection. This would explain why we did not detect a GRK2-phosphopeptide with our phosphoserine and-threonine specific protocol.

Substrates of GRK2

In the previous version of our manuscript we had a section on the role of HDAC6 as potential GRK2 target in the discussion but we appreciate the reviewer's point that we should expand on the potential targets of GRK2. The canonical target of GRK2 is beta-arrestin, which plays an important role in G protein-coupled receptor (GPCR) signalling. However, we did not detect any phosphopeptides of beta-arrestin in our proteomic analysis and we could not detect any change in phosphorylation status of beta-arrestin upon IAV infection when employing the Phos-tag method. Furthermore, a meta-analysis of several RNAi screens for host factors required for IAV infection predicted no significant impact of beta-arrestin knockdown on IAV infection. We thus conclude that the proviral effect of GRK2 is mediated by a non-canonical substrate of GRK2. Interestingly, HDAC6 has been described as substrate for GRK2 (3) and HDAC6 is also known to play a role in IAV uncoating (4). However, it was reported that phosphorylation of HDAC6 impacts its deacetylase activity, which was shown not be involved in HDAC6' role in uncoating. Furthermore, we did not detect any HDAC6 phosphopeptide and did not observe changes in HDAC6 phosphorylation upon IAV infection. We therefore believe that HDAC6 is unlikely to mediate the proviral effect of GRK2. We have included a section that discusses these results in l. 418-431.

In addition, we analysed the impact of GRK2 on the phosphorylation status of viral proteins. We again used Phos-tag SDS-PAGE (2) and analysed the banding pattern of viral proteins early in infection in the absence or presence of the GRK2 inhibitor GRK2i. Given that we found GRK2 to be required for IAV entry at the level of uncoating we focused on viral M1, M2 and NP, the major viral players in uncoating. As shown in the new supplementary figure S6 we did not detect phosphorylated forms of NP and M1 in control cells or inhibitor-treated cells. In line with published results (5), we detected phosphorylated M2 in the incoming virus but no difference in banding pattern was observed between control- and inhibitor-treated samples. We thus conclude that GRK2 does not impact the phosphorylation status of M1, M2 and NP and most likely does not exert its proviral role during uncoating via phosphorylation of viral proteins. These novel results are shown in supplementary figure S6, described, and discussed in l. 326-332 and l. 431-434.

Minor points:

1. Line 85, to my understanding, “the phosphorylation” should be “the tyrosine phosphorylation” of EGFR. If it is true, what is the rationale not to include tyrosine phosphorylation profile?

We changed “phosphorylation” to “tyrosine phosphorylation” in 1.88 according to the reviewer’s suggestion. It would certainly be very interesting to also look at tyrosine phosphorylation events in response to influenza virus infection. However, in contrast to serine and threonine phosphoproteomics, in which metal affinity purification is used for phosphopeptide enrichment, tyrosine phosphoproteomics mostly relies on the use of antibody-based immunoaffinity purification (6). Although the quality of the anti-phospho-Tyr antibodies and the number of available protocols have increased during the last years, this approach is still technically challenging and the coverage and number of phosphopeptides detected in such studies is modest when compared with similar ones focusing on serine and threonine phosphorylation.

2. Infection causes cell stress. Is there any cross-talk with the stress signaling pathway by network analysis?

According to the reviewer’s suggestion we performed additional network analysis to test if any crosstalk to stress-associated signaling pathways, such as FoxO signaling (KEGG identifier hsa04068), p53 signalling (hsa04115), autophagy (hsa0440) etc., could be revealed. No significant enrichment of any stress-related pathways was detected for our dataset indicating that the very early infection events do not cause detectable cell stress. We hypothesize that massive production of viral transcripts and proteins later in infection are the main triggers for stress pathway activation, in particular for metabolic and genomic stress.

Reviewer #3 (Remarks to the Author)

In this study a phosphoproteomic screen was performed with the aim to identify cellular kinases involved in influenza A virus infection. To this end, phosphoproteomic profiling was performed after 5 min and 15 min infection. Kinases activated in response to infection were identified via bioinformatics analysis. Inhibitors of several of the identified kinases were shown to inhibit infection. GRK2 was shown to be required for IAV uncoating. A SSRI that also inhibits GRK2 was shown to decrease IAV replication when administered prior to infection in mice. In general it is a very nice study that is well executed and written. However, there are a few issues, the authors should address.

1. It is not clear to me why the 5 and 15 min time points were chosen. The only reason seems to be the study that shows that at these time points EGFR phosphorylation was observed?

The goal of our study was to reveal signalling events induced by the very first steps of the infection cycle, in particular virus binding to cells. We hypothesized that these early events would already trigger signalling cascades that enable the following steps of the replication cycle. As it had been shown by Eierhoff et al. that HA binding to cells can trigger EGFR signalling (7) we used an assay to monitor EGFR activation after addition of influenza virus to cells. The results from this assay suggested

that 5 and 15 min p.i. would be suitable time points. As a similar study on early signalling events induced by HIV had also analysed time points in the range of 1-15 min (8) we decided to perform the phosphoproteomics analysis for these two time points. We expanded the explanation for our choice of time points in l. 85-93.

2. The moi used in the study mentioned in point 1 (ref 8) is very high. What is the moi (and viral strain) used in the phosphoproteomic screen in this study. How does this relate to the number of IAV particles per cells? How physiologically relevant are the moi used, and are similar phosphorylation profiles expected at lower moi?

Whereas the study by Eierhoff et al. used an MOI of 100 and avian IAV strain FPV to study the signalling events induced by early steps of virus infection (7) we used an MOI of 25 PFU per cell and strain A/WSN/33 for our phosphoproteomics analysis. In our revised version, we also included this information about our experimental set-up in the legend for figure 1 and in the results section (l. 91). In order to determine the number of virus particles that corresponds to an MOI of 25 we quantified the amount of viral M segment by RT-qPCR in our virus preparation. We included serial dilutions of an M-segment encoding plasmid as standard to obtain absolute copy numbers and found that an MOI of 25 PFU per cell corresponds to approximately 25.000 virus particles per cell. This may seem unphysiological at first glance and certainly does not reflect the initial infection event *in vivo*. However, when virus replication takes place in the respiratory epithelium, large amounts of virus are produced and epithelial cells can thus be exposed to high doses of virus secreted from neighbouring epithelial cells. Furthermore, the sensitivity of current phosphoproteomics approaches would not be sufficient to detect phosphorylation events that only occur in few cells during low MOI infections as often only a small proportion of a given protein becomes phosphorylated (see GRK2 phosphorylation in new figure 3e). Using an MOI of 25 PFU per cell resulted in a 94% infection rate in A549 cells. Further lowering of the MOI would thus compromise the sensitivity of our approach. Given these technical limitations and the possible physiological relevance of high MOI infections, we believe that our study yielded relevant information about signalling events induced early in IAV infection.

3. Is it possible that part of the phosphorylation signature observed does not result from IAV particles attaching to cells, but from other factors induced and released from infected cells (e.g. interferons or interferon-stimulated gene products). The virus stocks do not seem to be purified. Would it be possible to perform control experiments using conditioned media from which particles are removed?

In order to avoid contributions of interferons to the phosphorylation signature we used virus stocks that had been prepared in embryonated chicken eggs and used allantoic fluid from mock-infected eggs as control. If chicken interferon was present in our virus stock it would not be able to induce signalling on human cells due to the species-specific action of interferons. However, as noted by the reviewer we cannot exclude contributions from other components of the allantoic fluid. To address this point we performed a proteomic analysis of our lysates used for the phosphoproteomic analysis and identified chicken proteins present in the different samples. The results of the proteomic analysis are shown in the new supplementary table 1. Only two chicken proteins were at least 2-fold more

abundant in the sample infected for 5 min compared to the mock allantoic fluid and three proteins were more abundant in the sample infected for 15 min compared to mock. None of these proteins was present more abundantly at both time points and we can conclude that contributions from chicken proteins present in the virus stock are unlikely to have impacted our results. These new results are described in I. 98-105. Furthermore, we performed an additional control experiment, in which we tested for GRK2 activation upon infection with a purified virus stock. As shown in the new supplementary figure 4g, the purified virus stock also activated GRK2 and we can thus conclude that the activation of GRK2 is independent of allantoic fluid components other than viral particles. These new results are described in I. 256-259.

4. In the mouse experiment, the authors do not use the inhibitor that they used throughout the study, but rather a SSRI that also inhibits GRK2 activity. It cannot be excluded that the inhibition observed results from the drug acting as a SSRI rather than from inhibition of GRK2. Were animal studies also performed using the other GRK2 inhibitor and what was the outcome? The authors could perform inhibition experiment (at least in vitro) using another SSRI that not inhibits GRK2 (e.g. fluoxetine?) to show that SSRIs do not generally inhibit IAV infection and uncoating.

We tested only paroxetine as GRK2 inhibitor in mice as two studies had already demonstrated the effect of paroxetine on GRK2 activity in mice (9, 10). In contrast, no data were available for GRK2i use in mice and we therefore reasoned that paroxetine was the better choice for *in vivo* experiments (I. 337; 356-358). However, we do appreciate the reviewer's point that paroxetine could also impact IAV infection via its activity as serotonin reuptake inhibitor. Therefore, we now show in the new supplementary figure 7a that fluoxetine as representative SSRI does not inhibit IAV infection. We describe these results in I. 339-341.

References

1. Bancroft JD, Gamble M. Theory and Practice of Histological Techniques: Churchill Livingstone; 2008.
2. Kinoshita-Kikuta E, Aoki Y, Kinoshita E, Koike T. Label-free kinase profiling using phosphate affinity polyacrylamide gel electrophoresis. *Molecular & cellular proteomics* : MCP. 2007;6(2):356-66.
3. Lafarga V, Aymerich I, Tapia O, Mayor F, Jr., Penela P. A novel GRK2/HDAC6 interaction modulates cell spreading and motility. *EMBO J*. 2012;31(4):856-69.
4. Banerjee I, Miyake Y, Nobs SP, Schneider C, Horvath P, Kopf M, et al. Influenza A virus uses the aggresome processing machinery for host cell entry. *Science*. 2014;346(6208):473-7.
5. Hutchinson EC, Denham EM, Thomas B, Trudgian DC, Hester SS, Ridlova G, et al. Mapping the phosphoproteome of influenza A and B viruses by mass spectrometry. *PLoS Pathog*. 2012;8(11):e1002993.
6. van der Mijl JC, Labots M, Piersma SR, Pham TV, Knol JC, Broxterman HJ, et al. Evaluation of different phospho-tyrosine antibodies for label-free phosphoproteomics. *J Proteomics*. 2015;127(Pt B):259-63.
7. Eierhoff T, Hrincius ER, Rescher U, Ludwig S, Ehrhardt C. The epidermal growth factor receptor (EGFR) promotes uptake of influenza A viruses (IAV) into host cells. *PLoS Pathog*. 2010;6(9):e1001099.
8. Wojcechowskyj JA, Didigu CA, Lee JY, Parrish NF, Sinha R, Hahn BH, et al. Quantitative phosphoproteomics reveals extensive cellular reprogramming during HIV-1 entry. *Cell Host Microbe*. 2013;13(5):613-23.
9. Schumacher SM, Gao E, Zhu W, Chen X, Chuprun JK, Feldman AM, et al. Paroxetine-mediated GRK2 inhibition reverses cardiac dysfunction and remodeling after myocardial infarction. *Sci Transl Med*. 2015;7(277):277ra31.

10. Thal DM, Homan KT, Chen J, Wu EK, Hinkle PM, Huang ZM, et al. Paroxetine is a direct inhibitor of g protein-coupled receptor kinase 2 and increases myocardial contractility. *ACS Chem Biol.* 2012;7(11):1830-9.

REVIEWERS' COMMENTS:

Reviewer #1 (Remarks to the Author):

The authors have satisfactorily addressed my concerns so from my point of view, the manuscript can now be accepted for publication.

Reviewer #2 (Remarks to the Author):

The authors addressed my comments satisfactorily. I have no further concerns.

Reviewer #3 (Remarks to the Author):

The reviewers' comments were well addressed.